

# Seasonal variability and sources of *in situ* brGDGT production in a permanently stratified African crater lake

Loes G.J. van Bree[1], Francien Peterse[1], Allix J. Baxter[1], Wannes De Crop[2], Sigrid van Grinsven[3], Laura Villanueva[3], Dirk Verschuren[2], Jaap S. Sinninghe Damsté[1,3]

[1]Utrecht University, Department of Earth Sciences, Princetonlaan 8A, 3584 CD Utrecht, the Netherlands
[2]Ghent University, Limnology Unit, K.L. Ledeganksttraat 35, B-9000 Gent, Belgium
[3]NIOZ Royal Institute for Sea Research, Department of Marine Microbiology and Biogeochemistry, and Utrecht University, PO Box 59, 1790 AB Den Burg, the Netherlands

*Correspondence to*: Francien Peterse (f.peterse@uu.nl)

**Abstract.** Lake sediments are important archives of continental climate history, and their lipid biomarker content can be exploited to reconstruct paleoenvironmental conditions. Branched glycerol dialkyl glycerol tetraethers (brGDGTs) are bacterial membrane lipids widely used in paleoclimate studies to reconstruct past temperature. However, major gaps still exist in our understanding of the environmental controls on *in situ* (i.e., aquatic) production in lake systems. In Lake Chala, a permanently stratified tropical crater lake in East Africa, we determined the concentrations and fractional abundances of individual brGDGTs along depth profiles of suspended particulate matter (SPM) collected monthly from September 2013 to January 2015, and in settling particles collected monthly at 35 m water depth from August 2010 to January 2015, and compared these brGDGT distributions with those in surficial lake-bottom sediments and catchment soils. We find that brGDGTs are primarily produced within the water column, and that their concentrations and distributions vary greatly with depth and over time. Comparison with concentration-depth profiles of the monthly distribution and abundance of bacterial taxa, based on 16S rRNA gene amplicon sequencing and quantification, indicates that Acidobacteria are likely not the main producers of brGDGTs in Lake Chala. Shallowing of the oxic-anoxic boundary during seasonal episodes of strong water-column stratification promoted production of specific brGDGTs in the anoxic zone. BrGDGT distributions in the water column do not consistently relate with temperature, pH, or dissolved-oxygen concentration, but do respond to transitions between episodes of strong stratification and deep (but partial) lake mixing, as does the aquatic bacterial community. Hence, the general link between brGDGT distributions and temperature in brGDGT-based paleothermometry is more likely driven by a change in bacterial community composition than by membrane adaptation of specific members of the bacterial community to changing environmental conditions. Although temperature is not the principal driver of distributional changes in aquatic brGDGTs in this system, at least not during the 17-month study period, abundance-weighted and time-integrated averages of brGDGT fractional abundance in the 53-month time series of settling particles reveal systematic variability over longer time scales that indirectly relates to temperature. Thus, although we do not as yet fully understand the drivers of modern-day brGDGT fluxes and distributions in Lake Chala, our data do support the application of brGDGT paleothermometry to time-integrated archives such as sediments.





## 1.    Introduction

Lake sediments are important archives of continental climate history, especially in (sub-) tropical regions where
other long-term, high-resolution natural archives such as ice cores or speleothems are lacking. Lipid biomarkers
preserved in those sediments can be used to examine present and past environmental conditions, and often
provide more specific information on those conditions than bulk geochemical proxies (see Castañeda and
Schouten, 2011 for a review). For example, plant waxes stored in lake sediments are used to reconstruct past
vegetation and hydroclimate dynamics (e.g., Freeman and Pancost, 2013; Diefendorf and Freimuth, 2017), while
the presence and distribution of (iso-)loliolide, long-chain $n$-alk-1-enes or 1,15 $n$-alkyl diols can be linked to
shifts in algal community composition and/or primary productivity (e.g., Volkman et al., 1998; Castañeda and
Schouten, 2011; van Bree et al., 2018).
Temperature is probably the most important climate parameter to be reconstructed quantitatively from
lacustrine settings, but despite more than a decade of substantial effort this remains challenging. One promising
proxy for continental paleothermometry is based on a suite of membrane lipids supposed to be derived from
bacteria, namely the branched glycerol dialkyl glycerol tetraethers (brGDGTs; Sinninghe Damsté et al., 2000).
These consist of tetra- (I), penta- (II) or hexamethylated (III) components, with none (suffix a), one (b) or two (c)
cyclopentyl moieties, and with methyl groups on the $5^{th}$ (5-methyl) or $6^{th}$ (6-methyl; indicated with a prime
notation) carbon position of their alkyl chain (Fig. 1; Sinninghe Damsté et al., 2000; Weijers et al., 2006, De
Jonge et al., 2013). The distribution of brGDGTs in modern surface soils and peats shows empirical relationships
with mean annual air temperature (MAAT) and the pH of the soil or peat in which they are produced (Weijers et
al., 2007b; De Jonge et al., 2014a; Naafs et al., 2017a,b). Although the bacteria that produce brGDGTs are still
largely unknown (Sinninghe Damsté et al., 2018), this relationship has been commonly used as proxy for
continental air temperature in paleoclimate reconstructions. For example, analysis of brGDGTs in loess soils,
peats and marine sediments has produced paleotemperature records across a wide range of geological ages (e.g.,
Weijers et al., 2007a; Peterse et al., 2011; Naafs et al., 2017a; Zheng et al., 2017).
The application of this temperature proxy on lake-sediment records was initially based on the premise
that all sedimentary brGDGTs are derived from catchment soils, and washed into the lake by erosion. However,
when brGDGT distributions in lake sediments were found to differ substantially from those in soils surrounding
the lake, it became clear that there must be an additional, *in situ* source of brGDGTs contributing to the lake
sediments (e.g., Tierney and Russell, 2009; Tierney et al., 2009; Sinninghe Damsté et al., 2009; Loomis et al.,
2011; Schouten et al., 2013; Buckles et al., 2014; Colcord et al., 2015; Li et al., 2016). In addition, brGDGT
isomers of type IIIa with methyl branches at the $5^{th}$ position on the one end and at the $6^{th}$ position on the other
end (IIIa″), have so far been detected exclusively in lakes and not in soils, providing further evidence for their *in*
*situ* production (Weber et al., 2015, 2018). Furthermore, comparison of the stable carbon isotopic composition
($\delta^{13}$C) of brGDGTs in lakes and nearby soils indicates distinctive signatures for, and thus sources of lacustrine
and soil-derived brGDGTs, with the lacustrine brGDGTs being significantly more $^{13}$C-depleted (Weber et al.,
2015; 2018; Colcord et al., 2017).
Water-column studies show that brGDGT concentrations generally increase below the oxycline,
suggesting that they are mainly produced in the anoxic portion of the hypolimnion (Sinninghe Damsté et al.,
2009; Bechtel et al., 2010; Blaga et al., 2011; Woltering et al., 2012; Buckles et al., 2014; Loomis et al., 2014b;



Miller et al., 2018). Also brGDGT production often varies seasonally (Sinninghe Damsté et al., 2009; Woltering et al., 2012; Buckles et al., 2014), which may introduce a temperature bias towards the season(s) with high brGDGT production (Loomis et al., 2014b; Miller et al., 2018). The contribution of aquatic brGDGTs, especially that of IIIa, generally results in a substantial underestimation of present-day temperature when the transfer function based on soil brGDGTs is used (Tierney et al., 2010), which has stimulated the development of temperature calibrations based on lake sediments (Tierney et al., 2010; Pearson et al., 2011; Sun et al., 2011; Loomis et al., 2012; Russell et al., 2018). As in soils the amount and distribution of brGDGTs in lake sediments seems to be influenced mostly by temperature and lake-water pH (Tierney et al., 2010; Sun et al., 2011; Loomis et al., 2014a), although a wide range of other factors such as oxygen availability (e.g., Tierney et al., 2012; Loomis et al., 2014a; Weber et al., 2018), light and mixing regime (Loomis et al., 2014b), nutrients (Tierney et al., 2010; Loomis et al., 2014a), water chemistry including alkalinity (Schoon et al., 2013), redox state (Weber et al., 2018) and conductivity (Tierney et al., 2010) have also been suggested to influence the *in situ* production of brGDGTs in lakes.

Temperature calibrations based on brGDGTs in soils and peats have substantially improved following the identification and chromatographic separation of 5-methyl and 6-methyl brGDGT isomers (De Jonge et al., 2014a; Naafs et al., 2017a, 2017b; Dearing Crampton-Flood et al., 2020). Initial scanning of surficial bottom sediments from East African lakes revealed that especially 6-methyl brGDGTs behave differently in lakes compared to soils, suggesting that they are produced by different bacteria, or that brGDGT producers in lakes respond differently to environmental changes than those in soils (Russell et al., 2018). Separation of the 5-methyl and 6-methyl brGDGTs yields slightly better error statistics for the East African lake calibration and lacks outliers such as are present in the calibration without separation of these isomers, re-affirming the potential of brGDGTs for paleotemperature reconstructions in lakes (Russell et al., 2018). Nevertheless, Weber et al. (2018) recently showed that variations in brGDGT composition above and below the oxycline in Lake Lugano (Switzerland) are linked to the occurrence of distinct bacterial groups that thrive in the oxic and anoxic parts of the water column. In addition, the carbon isotopic composition of brGDGTs in the sediments of Alpine lakes indicates that brGDGT producers are differentiated according to lake trophic status. Together, this suggests that brGDGT signatures in a lake sediment record may also be influenced by temperature-independent factors, such as variations in community composition and primary production (Weber et al., 2018).

In this study we examined brGDGTs in suspended particles (SPM) from the water column of a permanently stratified lake (Lake Chala) in tropical Africa over a 17-month period to further constrain the seasonal and depth distribution of different brGDGTs, to identify their main producers, and to ascertain the sources of brGDGTs eventually stored in lake sediments. To this effect the SPM brGDGT data were compared with measurements of temperature, pH and dissolved oxygen (DO) obtained through concurrent water-column monitoring, and with the composition and abundance of bacterial taxa in the SPM based on 16S rRNA gene amplicon sequencing and quantification. We also analyzed brGDGTs in settling particles collected at monthly intervals over a 4.5-year (53-month) period, to reveal possible long(er)-term trends in the seasonality of brGDGT production in this lake, which may help elucidate its environmental drivers. Finally, comparison of the aquatic brGDGT signature with that of soils surrounding the lake and the lake sediments itself was expected to shed light on the significance of paleoclimate reconstructions based on brGDGTs in lake sediments.





**2.     Material and methods**
**2.1.   Study system**
Lake Chala (locally "Challa", after a nearby village) is a small (4.2 km$^2$), deep (~90 m) and permanently
stratified (meromictic) crater lake, situated at ~880 m above sea level and bridging the border of Kenya and
Tanzania (3°19'S, 37°42'E) in the foothills of Mt. Kilimanjaro. At this near-equatorial location, mean monthly
air temperature (MMAT) varies between 20-21 ℃ in July-August and 25-27 ℃ in January-February. The
tropical rain belt associated with latitudinal migration of the Inter-Tropical Convergence Zone (ITCZ) passes
across the region twice yearly, resulting in two wet seasons and two dry seasons. Short rains occur from late
October to December, and long rains from March to mid-May. The principal dry season occurs during southern
hemisphere winter (June-September) and is characterized by lower air temperature and higher wind speeds. The
latter drive evaporative cooling which promotes deep convective mixing of the water column of Lake Chala to
~40-60 m depth, while the deeper water remains permanently stratified and anoxic (Wolff et al., 2011; Buckles
et al., 2014; van Bree et al., 2018). A second period of lesser mixing, to 25-30 m depth, occurs during the short
dry season of January-February. Primary productivity is highest during the principal dry season (June to
October), when nutrient-rich deep water is mixed upwards into the normally unproductive epilimnion (Wolff et
al., 2014; van Bree et al., 2018). The lake's water balance is partly maintained by rainfall on the lake surface and
over the steep-sloping crater basin, occasionally supplemented by high rainfall over the catchment of a small
creek which breaches the north-western crater rim (Buckles et al., 2014). As lake-surface evaporation (1700 mm
yr$^{-1}$) greatly exceeds annual rainfall (600 mm yr$^{-1}$), water balance is maintained by substantial subsurface inflow
(Payne, 1970) of water that originates from percolation in or above the forest belt on Mt. Kilimanjaro (Hemp,
2006; Bodé et al., 2020).

**2.2.    Field observations and sample collection**
**2.2.1.   Temperature, pH, and dissolved-oxygen monitoring of the water column**
Vertical profiles of temperature, dissolved oxygen (DO), conductivity (K25) and pH were measured at 2-m
intervals through the upper 50 m of the water column using a Hydrolab Quanta® multi-sensor probe at a mid-
lake position (Fig. 2), at monthly intervals between September 2013 and January 2015 (van Bree et al., 2018).
Additionally, water temperature was measured by automatic temperature loggers, at 2-hourly intervals between
September 2010 and January 2015, suspended at a selection of the following water depths: 2, 10, 20, 25, 30, 35,
40, 45, 50 and 85 m. The set of monitoring depths varied over time due to the occasional malfunctioning and
subsequent replacement of loggers. Due to loss of logger data during retrieval, no water-column temperature
information is available for the period between 7 January and 11 September 2012. The entire 53-month
temperature record was corrected for drift of individual loggers, using the Hydrolab profiles as reference. Periods
of water-column mixing and stratification were determined on the basis of the temperature-logger time series, or
estimated on the basis of mean monthly air temperature (MMAT) data from Buckles et al. (2014) for the
abovementioned hiatus period.

**2.2.2.   Suspended particulate matter sampling**
Collection of the SPM profiles used in this study has been described by van Bree et al. (2018). In short, 5 to 10 L
of lake water was collected at 13 discrete depths, monthly between September 2013 and January 2015. The



samples were filtered on pre-combusted glass fiber GF/F filters (142 mm diameter, Whatman), stored frozen,
and freeze-dried prior to analysis. The SPM was collected at or near the start of every month as discussed here,
with the sample taken at, for example, 07-09-2013 representing September 2013, and the sample taken at 30-09-
2013 representing October 2013 (see Table S.1).

**2.2.3.    Sampling of settling particles**
A sediment trap (UWITEC, double-funneled, 86 mm diameter) suspended in 35 m water depth at a mid-lake
position (Fig. 2) was installed in November 2006, after which it was emptied and redeployed at about monthly
intervals (Table S1). The collected material was allowed to settle for two days, and stored frozen after
decantation of excess water. Prior to analysis, the samples were thawed, filtered over pre-weighed and pre-
combusted (400ºC, 5 h) glass fiber GF/F filters (110 mm diameter, Whatman), then frozen and freeze-dried.
Bulk mass flux was calculated for each month by using the dry weight of the collected particles, the number of
days covered and the surface area of the sediment trap (58 cm$^2$), and is expressed as mg m$^{-2}$ day$^{-1}$. This study
focuses on the brGDGTs in settling particles representing the period from September 2010 until January 2015 ($n$
= 53; Table S.2).

**2.2.4.    Soil sampling**
Seven soil samples from the collection obtained by Buckles et al. (2014) were selected for brGDGT analysis
based on site dissimilarity, i.e. from different origins (lakeshore forest, crater rim, savanna hinterland, small
ravine; see Table S3) as described in the original study.

**2.2.5.    Lake sediment sampling**
Intact surficial lake-bottom sediment (2-5 cm depth) from 3 sites (CH10-06G: 3°19.049'S, 37°41.879'E; CH10-
09G: 3°18.704'S, 37°41.448'E and CH10-10G: 3°18.575'S, 37°41.419'E; see Table S4) forming a transect from
close to the creek inlet towards the middle of the lake (Buckles et al., 2014) was collected by gravity coring in
January-February 2010, then freeze-dried and homogenized prior to extraction.

**2.3.    Sample preparation and lipid extraction**
Sample preparation for SPM was described in detail by van Bree et al. (2018). For this study, SPM was used
from all depths for the months of November 2013 and August 2014, as well as from 0, 10, 25, 35, 50, 60, 70 and
80 m depth for all other months (total $n$ = 146). In short, the freeze-dried filters were cut in small pieces and
extracted using a modified Bligh-Dyer method. Each extract was acid-hydrolyzed with 1.5 N HCl in methanol
(MeOH). After a 2 h reflux at 80 ºC, the pH of the hydrolyzed extract was adjusted to 4-5 by addition of 1 N
KOH / MeOH (96%), and washed three times with dichloromethane (DCM). The combined organic phases were
passed over a Na$_2$SO$_4$ column and dried under N$_2$. The total lipid extract (TLE) obtained was separated on an
activated Al$_2$O$_3$ column into an apolar, neutral and polar fraction, using hexane:DCM (9:1, v/v), DCM, and
DCM:MeOH (1:1, v/v) as eluents, respectively. The freeze-dried filters with sediment-trap material were cut in
small pieces and extracted directly by acid hydrolysis. The then obtained TLE was further processed similar to
the SPM TLE. Also the lake sediment samples were extracted and processed in similar fashion as the SPM.





A known amount of internal standard (99 ng $C_{46}$ GDGT; Huguet et al., 2006) was added to the polar
fraction of SPM, settling particles and sediments. All polar fractions of SPM, sediment trap, surface sediments
and soils were re-dissolved in hexane:isopropanol (99:1, v/v) and passed over a 0.45 µm PTFE filter.

**2.4.    GDGT analysis and proxy calculation**
GDGT analysis was performed with an Agilent 1260 Infinity ultrahigh performance liquid chromatography
(UHPLC) coupled to an Agilent 6130 single quadrupole mass detector, either at Utrecht University (most SPM,
soil, surface sediments) or at the NIOZ (settling particles, SPM at 0 m, except November 2013 and September
2014) following the method of Hopmans et al. (2016). Separation was achieved by two silica Waters Acquity
UPLC HEB Hilic (Ø1.7 µm) columns at 30 °C, preceded by a guard column with similar packing. Isocratic
elution was used for GDGT separation, starting with 82% A (hexane) and 18% B (hexane : isopropanol, 9:1) for
25 min at a flow rate of 0.2 mL min$^{-1}$, followed by a linear gradient to 70% A and 30% B for 25 min. Injection
volume was 10 µL for settling particles, sediment and soils, and 20 µL for SPM. Ionization of the GDGTs was
achieved by atmospheric pressure chemical ionization with gas temperature of 200 °C, vaporizer temperature of
400 °C, $N_2$ flow of 6 L min$^{-1}$, capillary voltage of 3500 V, nebulizer pressure of 25 psi and corona current of 5.0
µA as source conditions.
GDGTs were identified by detecting the $[M+H]^+$ ions in selected ion monitoring (SIM) mode for *m/z*
1018, 1020, 1022, 1032, 1034, 1036, 1046, 1048, 1050 (brGDGTs) and 744 (internal $C_{46}$ GDGT standard). Peak
area integration of the GDGTs was done with Chemstation (SPM, soil, sediment) or Agilent Masshunter (settling
particles, SPM at 0, 35, 60 and 70 m) software. For quantification, areas were compared to that of the internal
standard, assuming a comparable response of the mass spectrometer for all GDGTs. Fractional abundances of
brGDGTs were calculated by dividing the peak area of a specific brGDGT divided by the peak areas of all
measured brGDGTs.
The Roman numerals in the following equations refer to the molecular structures of GDGTs as shown
in Fig. 1, with 6-methyl brGDGTs distinguished by an accent, and square brackets indicating the fractional
abundances of the 15 different brGDGTs. The Cyclisation of Branched Tetraethers (CBT′) was defined by De
Jonge et al. (2014b) as

CBT′= –log {([Ic] + [IIa′] + [IIb′] + [IIc′] + [IIIa′] + [IIIb′] + [IIIc′]) / ([Ia] + [IIa] + [IIIa])}         (1)
where [x] refers to the fractional abundance of a specific brGDGT.

The isomerization ratio of the 6-methyl penta- and hexamethylated brGDGTs over 5-methyl and 6-methyl
brGDGTs ($IR_{6ME}$) was modified from De Jonge et al. (2014b) and Sinninghe Damsté (2016), and calculated as

$IR_{6ME}$ = ([IIa′] + [IIb′] + [IIc′] + [IIIa′] + [IIIb′] + [IIIc′]) / ([IIa] + [IIb] + [IIc] + [IIIa] + [IIIb] + [IIIc] + [IIa′] +
[IIb′] + [IIc′] + [IIIa′] + [IIIb′] + [IIIc′])                                                         (2)

Mean annual air temperature (MAAT) was reconstructed with the stepwise-forward-selection (SFS) calibration
of the brGDGT distribution in the East African lakes dataset (Russell et al., 2018):



$MAAT_{SFS} = 23.81 - 31.02*[IIIa] - 41.91*[IIb] - 51.59*[IIb'] - 24.70*[IIa] + 68.80*[Ib]$          (3)

Surface-water pH was reconstructed with the Russell et al. (2018) transfer function determined for East African
lakes:

Surface water pH = $8.95 + 2.65*CBT'$          (4)

**2.5.    Determination of 16S rRNA gene diversity and abundance**
DNA was extracted from 1/32 section of the SPM filters using the PowerSoil DNA extraction kit (Mo Bio
Laboratories, Carlsbad, CA, USA). The 16S rRNA gene amplicon sequencing and analysis was performed with
the general 16S rRNA archaeal and bacterial primer pair 515F and 806RB targeting the V4 region (Caporaso et
al., 2012), as described in Besseling et al. (2018). PCR products were gel-purified using the QIAquick Gel-
Purification kit (Qiagen), pooled and diluted. Sequencing was performed at the Utrecht Sequencing Facility
(Utrecht, the Netherlands) using an Illumina MiSeq 2x300 bp sequencing platform. The 16S rRNA gene
amplicon sequences were analyzed by an in-house pipeline including quality assessment by FastQC (Andrews,
2010), assembly of the paired-end reads with PEAR (Zhang et al., 2013), and taxonomic assignment (including
picking of a representative set of sequences with the 'longest' method; Caporaso et al., 2010) with BLAST
(Altschul et al., 1990) by using the Silva 128 release as reference database (https://www.arb-silva.de/). The 16S
rRNA gene copies were quantified using quantitative PCR (qPCR) with the same primer pair (515F, 806RB) as
used for amplicon sequencing. The qPCR reaction mixture (25 μl) contained 1 U of Pico Maxx high fidelity
DNA polymerase (Stratagene, Agilent Technologies, Santa Clara, CA), 2.5 μl of 10x Pico Maxx PCR buffer, 2.5
μl 2.5 mM of each dNTP, 0.5 μl BSA (20 mg/ml), 0.02 pmol/μl of primers, 10,000 times diluted SYBR Green®
(Invitrogen) (optimized concentration), 0.5 μl of $MgCl_2$ (50 mM), and ultrapure sterile water. The cycling
conditions for the qPCR reaction were the following: initial denaturation at 98 °C for 30 s, 45 cycles at 98 °C for
10 s, at 56 °C for 20 s, followed by a plate read, at 72 °C of 30 s and at 80 °C for 25 s. Specificity of the reaction
was tested with a gradient melting-temperature assay, from 55 °C to 95 °C with a 0.5 °C increment for 5 s. The
qPCR reactions were performed in triplicate with standard curves from 100 to 107 molecules per microliter.
qPCR efficiency for the 16S rRNA quantification was on average 95% with $R^2$=0.998.

**2.6  Statistical analysis**
To assess variability in brGDGT distribution among (types of) samples we performed principal component
analysis (PCA) in the R-package FactoMineR (Lê et al., 2008). For SPM, statistic analysis only used the
fractional abundance of the most abundant brGDGTs, i.e., Ia, Ib, IIa, IIa', IIb, IIb', IIIa and IIIa'. Water
temperature and pH were also included in the PCA, with pH between 50 and 90 m water depth assumed to be
similar to the pH measured at 50 m depth. Although complete pH profiles from Lake Chala show that pH still
decreases slightly with depth below 50 m (~0.5 pH units; Wolff et al., 2014), this represents only a quarter of the
total pH depth gradient.

Concentrations of brGDGTs (ng $L^{-1}$) were correlated with the estimated abundance of microbial groups

to assign a possible source of the former. The abundance of specific bacterial groups was estimated by
multiplying their relative abundance as obtained by 16S rRNA gene amplicon sequencing analysis with the



absolute abundance of microorganisms in a given sample based on qPCR. For simplicity it was assumed that
each microbe contains a single 16S rRNA copy in their genome; the abundance was accordingly expressed as
16S rRNA gene copies $L^{-1}$. On the premise that potential brGDGT producers must be frequently present in the
water column, microbial species present in less than 10% of the SPM samples were excluded from this
comparison.

**3. Results**
**3.1. Seasonal mixing and stratification**
Surface-water temperature as measured by temperature loggers at 2 m depth, over the 29-month period from
September 2012 to January 2015, ranged between 22.8 ºC during the mixing-season in August 2013 and 27.6 ºC
during the period of strong stratification in April 2013 (Fig. 3). Temperatures at 10, 20 and 25 m depth, i.e. in
lower epilimnetic and upper hypolimnetic water, varied seasonally with minima during the period of shallow
mixing (SM; January-February) and towards the end of the period of deep mixing (DM; May to mid-September).
Seasonal temperature variation at 50 m depth, i.e. near the mixing limit, was already strongly muted, and at 85 m
depth water temperature remained stable at ~22.4 ºC (Fig. 3), indicating lack of mixing. Over the 4.5-year
monitoring period from September 2010 to January 2015 also the upper water column of Lake Chala developed
stratification generally from September until April, with most strongly stratified conditions (i.e., greatest
temperature contrast between the surface and deep water) shortly after the annual peak in local air temperature
(February-March; Fig. 3).
During the 17-month period of lake monitoring between September 2013 and January 2015, the
thickness of oxygenated upper part of the water column, as based on the depth to anoxia (shallowest depth with
<0.2 mg $L^{-1}$ dissolved oxygen), varied between 17 m in October-November 2013 and 44 m in October-
November 2014 (Fig. 3). Depression of the oxycline resulted from convection-driven oxygen injection, mainly
towards the end of the stratified period and throughout the principal mixing period. In contrast, the period of
shallow mixing in January-February had little impact on the depth of the oxycline (Fig. 3).

**3.2. Spatial and temporal distribution of brGDGTs in SPM**
BrGDGTs were detected in all SPM samples analyzed (n = 143, Fig. 4). However, the abundance of brGDGTs
with one or two cyclopentyl moieties (types b and c; Fig. 1) was often too low for reliable quantification (i.e.,
peak height less than three times the noise level of the baseline). Specifically, concentrations of brGDGTs IIIc
and IIIc′ were always below detection limit; and brGDGTs Ic, IIc, IIc′, IIIb and IIIb′ were present in less than
half of the SPM samples and in very modest amounts, often around the detection limit. The IIIa″ isomer (Weber
et al., 2015), which has so far been detected only in lakes, was not detected at all in our samples. Consequently,
in the following analysis we focus on the eight brGDGTs that were detected in at least 60% of the samples (i.e.,
Ia, Ib, IIa, IIa′, IIb, IIb′, IIIa and IIIa′) unless stated otherwise.
The total concentration of these eight brGDGTs in the water column (ΣbrGDGTs) varied between 0.2
and 24 ng $L^{-1}$ (n = 143), and generally increased with depth, especially in the anoxic part of the water column
(Fig. 4). The concentration-weighted mean fractional abundances of the individual brGDGTs in SPM collected
at all depths above the sediment trap (0-35 m; SPM$_{abovetrap}$) and below it (40-90 m; SPM$_{belowtrap}$) over the 17-
month sampling period are shown in Fig. 5A. Pentamethylated (type II) brGDGTs were the most common





overall, with a summed fractional abundance ranging from 0.44 to 0.74 in the total dataset ($n = 143$) and average
values of 0.64 in $SPM_{abovetrap}$ ($n = 72$) and 0.72 in $SPM_{belowtrap}$ ($n = 71$). BrGDGT IIa′ was often dominant in the
SPM of Lake Chala, with a fractional abundance ranging from 0.11 to 0.60, and average values for $SPM_{abovetrap}$
and $SPM_{belowtrap}$ of 0.56 and 0.57, respectively (Fig. 5A). The tetramethylated (type I) brGDGTs amount to
between 0.06 and 0.47 of the brGDGT fractional abundance ($n = 143$), with average values of 0.31 in
$SPM_{abovetrap}$) and 0.21 in $SPM_{belowtrap}$; brGDGT Ia is the second-most abundant of all brGDGTs (Fig. 5A).
Finally, the concentration of hexamethylated (type III) brGDGTs is relatively variable with a combined
fractional abundance between 0 and 0.25 ($n = 143$), where brGDGT IIIa′ is the most dominant compound. The
hexamethylated brGDGTs have relatively low concentration-weighed average fractional abundances of 0.05 in
$SPM_{abovetrap}$ and 0.08 in $SPM_{belowtrap}$. Overall the tetramethylated brGDGTs are relatively more common in the
upper water column (fractional abundances $SPM_{abovetrap} > SPM_{belowtrap}$), whereas penta- and hexamethylated
brGDGTs are relatively more abundant in the lower water column ($SPM_{abovetrap} < SPM_{belowtrap}$).

In >86% of all SPM samples, 6-methyl brGDGTs were more abundant than 5-methyl brGDGTs, which

is reflected in an average $IR_{6ME}$ of 0.74 (range 0.38-1.00). The 5-methyl brGDGTs were relatively abundant only
between November 2013 and August 2014, with a maximal fractional abundance of 0.33 at 35 m in February
2014. Absolute concentrations of 5-methyl brGDGTs peaked at 8.0 ng $L^{-1}$ at 60 m depth in April 2014. This is
mainly the result of brGDGT-IIb, which contributed 4.8 ng $L^{-1}$ to this amount (Fig. 4). The concentration of 6-
methyl brGDGTs was typically highest in the non-mixing deepest part of the water column (>60 m), and reaches
9.3 ng $L^{-1}$ at 80 m in February 2014 (Fig. 4). However, averaged over the sampling period the concentration-
weighted fractional abundance of 6-methyl brGDGTs was quite similar in shallow and deep water, with an
$SPM_{abovetrap}$ value of 0.62 and $SPM_{belowtrap}$ value of 0.66.

The concentrations of both ΣbrGDGT and individual brGDGTs were highest in the anoxic part of the

Chala water column under stratified conditions (Fig. 4). Importantly, depth-integrated concentrations (i.e.,
averaged over the entire water column) were also highest during the stratification periods, and lowest towards
the end of the deep-mixing period in 2014 when the oxycline was maximally depressed. Towards the end of both
2013 and 2014, brGDGT concentrations increased when stratification developed after the period of deep mixing.
However, the two stratification periods differed with regard to the fractional abundance of individual brGDGTs.
During stratification in 2013/2014, the brGDGT assemblage mainly consisted of Ib and IIb at 35-60 m water
depth, and IIa′ and IIIa′ at 60-80 m, whereas during stratification in late 2014 and early 2015, concentrations of
Ib and IIb were strongly reduced and notably high concentrations of IIa′ and IIIa′ were evident up to 25-35 m
water depth (Fig. 4).

The first three principal components (PCs) of a principle component analysis (PCA) on the fractional

abundances of the eight major brGDGTs in all SPM samples ($n = 143$) together explain 83.5% of the observed
variation in their distribution (Figs. 6A-B). PC1 explains 49.4% of the variance, has strong negative loadings for
the 6-methyl brGDGTs IIa′ and IIIa′, and strong positive loadings for the 5-methyl brGDGTs Ib, IIa, IIb and IIIa.
PC2 explains 21.0% of the variance, and mainly shows strong negative loadings for Ia and IIb′. PC3 explains
13.1% of the variance and shows a strong positive loading for IIb′ and IIIa.






### 3.3. BrGDGTs in settling particles

The total brGDGT flux captured by the sediment trap at 35 m depth varied by two orders of magnitude (between 84 and 5300 ng m$^{-2}$ day$^{-1}$) over the 53-month period of sediment-trap deployment (Fig. 7B). Total brGDGT flux is not related (R$^2$ = 0.02, $p$ = 0.91) to the bulk flux of settling particles (Fig. 7A), nor does its production (or sedimentation) appear to be concentrated in a specific season. BrGDGT concentrations in the monthly collection of settled particles were generally higher than in the snap-shot SPM samples, enabling quantification of all studied brGDGTs except IIIc. Nevertheless, like in SPM, the fractional abundance of brGDGTs IIc, IIIb and IIIc′ was <0.02 at all times, and rarely >0.02 for Ic, IIc′, IIIa and IIIb′ (Fig. 5A). BrGDGTs IIc, IIIb and IIIc′ were also found in only 62-77% of the 53 trap samples, whereas all other brGDGTs were found in at least 94% of these samples.

The distribution of brGDGTs shows large variation throughout the period of settling particle collection (Figs. 7B-F). The majority of brGDGTs in settling particles were pentamethylated, with a combined fractional abundance ranging between 0.46 and 0.61 (n = 53), similar to what was found for SPM. BrGDGT IIa′ was again most often the dominant compound (fluxes of up to 3200 ng m$^{-2}$ day$^{-1}$ in November 2014; Fig. 5A), although at times IIb (340 ng m$^{-2}$ day$^{-1}$ in March 2014) and IIb′ (530 ng m$^{-2}$ day$^{-1}$ in August 2013) were also abundant in the settling particle flux. The fractional abundance of tetramethylated brGDGTs (mostly Ia, as in the SPM) ranged from 0.11 to 0.46, and hexamethylated brGDGTs (mostly IIIa′, as in the SPM) ranged from 0.07 to 0.28. Further, 6-methyl brGDGTs most often (41 out of 53 months) comprised at least 80% of the total 5- and 6-methyl brGDGTs (IR$_{6ME}$ >0.8), except in June 2011 and from November 2013 to September 2014 (Fig. 7C).

The first three PCs of the PCA on the fractional abundances of the eight brGDGTs most common in settling particles (Ia, Ib, IIa, IIa′, IIb, IIb′, IIIa, and IIIa′, as in the SPM) together explain 91.4% of the observed variation through time ($n$ = 53; Fig. 6C-D). PC1 explains 57.5% of the variance and has strong negative loadings for the 6-methyl brGDGTs IIa′ and IIIa′, and positive loadings for the 5-methyl brGDGTs Ia, Ib, IIa, IIb and IIIa. PC2 explains 23.7% of the total variance and has strong negative loadings for Ia and IIb′, and positive loadings for especially IIa, IIb and IIIa. PC3 explains 10.2% of the total variance and has a strong positive loading for Ia. Thus, variation in individual brGDGT distributions is overall similar in SPM and settling particles (cf. Figs. 6A-B and 6C-D).

The combined PCA on all 143 SPM and 53 sediment-trap samples (Fig. 6E) indicates that variation in brGDGT distributions is mainly structured by the relative abundance of 5- and 6-methyl brGDGTs (PC1 explains 48.6% of the total variance), and by a different behavior of brGDGTs Ia and IIb′ from the six other brGDGTs (PC2 explains 22.4% of the total variance). As expected, the brGDGT distribution in settling particles is most similar to that in SPM from the upper water column, i.e. sampled at depths situated above the sediment trap (Fig. 6E).

### 3.4. BrGDGTs in catchment soils

BrGDGTs in soils surrounding Lake Chala ($n$ = 7) are predominantly tetramethylated (fractional abundance 0.48-0.84), followed by pentamethylated (0.15–0.44) and hexamethylated (0.01–0.09) compounds (Fig. 5B). Tetramethylated brGDGTs as well as compounds IIa′, IIb′, and IIIa′ were present in all analyzed soils. Several penta- and hexamethylated brGDGTs were below detection limit in two (IIa, IIIb), three (IIb), four (IIa′, IIIb′), five (IIIa) or all seven (IIIc, IIIc′) soil samples, and the fractional abundance of IIc, IIc′, IIIb and IIIb′ was usually





<0.02 (Fig. 5A). The $IR_{6ME}$ of Chala soils ranges between 0.52 and 0.90, with an average value of 0.73 (Fig. 7C).
Variation in brGDGT distributions among soils is explained mainly by their location in- or outside of the crater
basin (hinterland savanna, ravine, crater rim or lakeshore forest), in line with results of earlier analyses that did
not differentiate between 5- and 6-methyl brGDGTs (Buckles et al., 2014). The brGDGT distributions in soils
differ substantially from those in SPM and settling particles, mainly because of a higher fractional abundance of
Ia, Ib and IIa, and lower proportion of 6-methyl brGDGTs (Fig. 5A). When soil brGDGT distributions are added
to the PCA of water column brGDGTs as passive samples, all soils plot in the third quadrant of positive PC1 and
negative PC2 values, distinct from the lake SPM and settling particles (Fig. 6E).

**3.5.    BrGDGTs in surficial lake-bottom sediments**
All brGDGTs except IIIc and IIIc′ were detected in recently deposited lake-bottom sediments ($n$ = 3), although
the fractional abundances of Ic, IIc, IIc′, IIIb, IIIb′ are always <0.02 (Fig. 5A). The distribution of individual
brGDGTs is highly similar among the three analyzed samples (Figs. 5B and 6E), with fractional abundances of
0.47-0.48 for penta-, 0.40-0.41 for tetra-, and 0.12 for hexamethylated brGDGTs, and Ia (0.31) and IIa′ (0.27)
being the dominant compounds. $IR_{6ME}$ is ~0.67 (Fig. 7C). The brGDGT distribution in these lake sediments falls
within the range of those found in SPM and settling particles (Fig. 5B), however with only positive PC1 scores
(Fig. 6E).

**3.6.    Microbial diversity and abundance in the water column of Lake Chala**
The diversity and abundance of prokaryotes in the water column of Lake Chala was determined by analysis of all
collected SPM samples ($n$= 216), using Q-PCR and 16S rRNA gene amplicon sequencing. Proteobacteria
formed the most important group of microbes but Acidobacteria, Actinobacteria, Chlorobi, Chloroflexi,
Cyanobacteria, Firmicutes, Bacteriodetes, Planctomycetes, Parcubacteria, and Verrucomicrobia, were also
present in varying relative amounts. The relative abundance of the Acidobacteria was 4% at maximum but on
average only represented 0.1% of the prokaryotic population (Table S5). Among the Acidobacteria, sequences
closely affiliated to Blastocatellia (subdivision (SD) 4), as well as those closely related to SD 21 and SD 6
dominated throughout the water column (Table S5).

Correlations between absolute concentrations of the eight most common brGDGTs in the Lake Chala

SPM and the spatiotemporal distribution of specific bacterial groups based on 16S rRNA gene abundance
estimates is shown in Fig. 8. For Acidobacteria, a suspected phylum of brGDGT producers (Sinninghe Damsté et
al, 2011), the highest degree of correlation was found between Acidobacteria SD 21 and brGDGTs Ib ($R^2$ = 0.23,
$p$ < 0.001, $n$ = 132) and IIb ($R^2$ = 0.22, p < 0.001, n = 117). However, most correlations between individual
brGDGTs and Acidobacteria SDs are weak ($R^2$ < 0.15; Fig. 8A). Outside the phylum Acidobacteria, modest
positive correlations ($R^2 \geq 0.2$) were found between at least one of the eight major brGDGTs and the 16S rRNA
gene abundance of 12 individual bacterial taxa (Fig. 8B). The highest positive correlation was found between
brGDGT IIa and an uncultured bacterium of the phylum Aminicenantes ($R^2$ = 0.40, $p$ < 0.001, $n$ = 125).

**4.    Discussion**
**4.1.    An aquatic origin of brGDGTs in Lake Chala**



BrGDGTs in lakes can originate from both terrestrial and aquatic sources, and hence a mixed signal can be
expected. There are several indications that the SPM of Lake Chala primarily contains brGDGTs produced
within the water column rather than being washed in with eroding catchment soils. Firstly, at all times in the
seasonal cycle brGDGT concentrations in the SPM show an order-of-magnitude increase with depth (Fig. 4).
Based on data from a limited number of SPM profiles, it was previously thought that this pattern mainly
originated because of favorable conditions for organic preservation in the anoxic lower water column of Lake
Chala (Sinninghe Damsté et al., 2009; Buckles et al., 2014). Such a presumed stable brGDGT reservoir might be
formed when slowly sinking organic particles become neutrally buoyant in the cooler hypolimnion and
consequently accumulate over time, combined with a lack of processes (such as grazing and aggregation) to
remove these particles from the water column (Sinninghe Damsté et al., 2009; Buckles et al., 2014). However,
since our data also show significant variation in the distribution of individual brGDGTs between different depth
intervals within the anoxic portion of the water column (Fig. 4), the concept of a static hypolimnetic brGDGT
reservoir is untenable. Secondly, the depth-integrated total brGDGT concentration in SPM is lower at the end of
the mixing season (and start of the ensuing stratification) than during peak stratification conditions (Fig. 4),
arguing against the notion that upwelling during the mixing season merely disperses deep-water brGDGTs
throughout the water column. Thirdly, the distribution and abundance of individual brGDGTs changes not only
with depth but also through time (Fig. 4). Especially the changes in the lower water column are remarkable given
the fact that the maximum mixing depth between September 2013 and January 2015 was limited to ~45 m (Fig.
3; van Bree et al., 2018). For example, the total brGDGT concentration at 80 m depth fluctuated between 0.98 ng
$L^{-1}$ (February 2014) and 16 ng $L^{-1}$ (October 2014), and the different temporal trends of the individual brGDGTs
result in variations in the degrees of cyclization, methylation, and 5- or 6-methyl positioning within the
brGDGTs. Finally, the contrast in brGDGT distributions between SPM from either the oxygenated or anoxic
parts of the water column (largely corresponding with the zones above and below the sediment trap; Fig. 4) on
the one hand, and soils on the other (Fig. 5A-B) strongly suggests that high deep-water brGDGT concentrations
do not result primarily from the accumulation of soil-derived brGDGTs preserved in anoxic conditions, but from
*in situ* production, especially below the oxycline. Our combined evidence indicates that over the studied 17-
month interval, (almost) all brGDGTs in the SPM of the water column of Lake Chala have an aquatic source
while terrestrial input is negligible. This result corroborates the findings of Buckles et al. (2014), and is also
consistent with the general lack of terrestrial biomarkers, such as long-chain *n*-alkanes, in the SPM of Lake
Chala during this same time interval (van Bree et al., 2018).

**4.2.    Spatiotemporal variation in brGDGT distributions**
There is large variation in the fractional abundances of brGDGTs in the water column of Lake Chala over time.
Under stratified conditions from November 2013 to August 2014, the expanded anoxic zone is characterized by
high fractional abundances of brGDGTs Ib and IIb, which both peak in abundance at 60 m depth (Fig. 4).
Although seemingly similar environmental conditions occurred at the end of 2014, when after the end of
seasonal deep mixing the oxycline moved upwards again, the concentrations of Ib and IIb do not return to their
earlier levels. Instead, concentrations of IIa′ and IIIa′ rapidly increase at this time. Hence, it appears that deeper
mixing promotes either the production of Ib+IIb (5-methyl brGDGTs with rings), as observed in late 2013 and
early 2014, or of IIa′+IIIa′ (6-methyl brGDGTs with no rings but additional methylation), as observed in late





2014. This temporal variation in brGDGT composition is captured by PC1 in the PCA, which clearly separates
5- and 6-methyl brGDGTs (PC1 49.4%; Fig. 6A). Moreover, the associated alternation of brGDGTs with and
without cyclopentyl moieties is reflected by PC2 (Fig. 6A-B). Notably, where 5-methyl brGDGTs mostly occur
between 35-60 m depth, 6-methyl brGDGTs generally reside in the lowermost portion of the water column (60-
80 m; Fig. 4). Also 5- and 6-methyl brGDGTs with cyclopentyl moieties (Ib and IIb vs IIb') occur in different
parts of the water column (Fig. 4). Although the concentrations of IIb' in SPM are overall quite low, this depth
segregation suggests that the incorporation of cyclopentyl moieties into 5-methyl and 6-methyl brGDGTs is
driven by different factors.

**4.3.    Membrane plasticity *vs* community changes in aquatic brGDGT producers**
It is generally assumed that brGDGT producers adjust the molecular structure of their membrane lipids in
response to environmental changes, and in fact, these membrane adaptations are at the heart of brGDGT-based
paleoenvironmental proxies (e.g. Weijers et al., 2007b). In general, the distribution of individual brGDGTs in the
SPM of Lake Chala is in line with the ambient environmental conditions. Overall dominance of the 6-methyl
brGDGTs (Fig. 5) may be associated with the relatively high pH of the lake (8.2-9.3 at the surface; Wolff et al.,
2014), given that 6-methyl brGDGTs are predominantly produced under high-pH conditions in soils (De Jonge
et al., 2014a) as well as river (De Jonge et al., 2014b) and lake (Russell et al., 2018) water. Also the relatively
low abundance of hexamethylated (both 5- and 6-methyl) brGDGTs is characteristic for warm lakes (Tierney et
al., 2010; Loomis et al., 2014a; Russell et al., 2018). However, in Lake Chala a straightforward link between
brGDGT distributions and the seasonal cycle of environmental conditions is not apparent, as illustrated by the
distinct brGDGT distributions in the two episodes of strong stratification during our 17-month study period, and
by the apparently different drivers of cyclization in 5-methyl versus 6-methyl brGDGTs (see section 4.2). The
fact that temporal variation in the fractional abundance of aquatically produced brGDGTs is not easily linked to
the seasonal cycle in either temperature, dissolved oxygen distribution, or pH suggests that the brGDGT
molecular structure is not primarily governed by membrane adaptation to changing abiotic conditions. Instead,
they may result from variation in the composition of the lake's bacterial community, which, if the different
bacterial taxa produce different brGDGTs, will result in different brGDGT assemblages at different times. This
was also observed in the deep and meromictic Lake Lugano (Switzerland), where compositional changes in
brGDGTs with depth are strongly related to bacterial community changes across the oxycline (Weber et al.,

2018).

The producers of aquatic brGDGTs in Lake Chala can potentially be identified by comparing the depth
distribution and temporal variation of individual brGDGTs with 16S rRNA gene data obtained from the same
SPM samples, following the approach of Weber et al. (2018) and Sollai et al. (2019). The only mesophilic
bacteria currently known to produce the assumed precursor lipids for brGDGTs, namely *iso*-diabolic acid and its
5- and 6-methylated derivatives, are Acidobacteria (Sinninghe Damsté et al., 2011a, 2018). However, their
presence has only been demonstrated in soil-derived aerobic Acidobacterial strains belonging to SDs 1, 3, 4, and
6, while strains of SDs 8, 10 and 23 do not contain these lipids. Ether-bound *iso*-diabolic acid and its derivatives
occur only in high abundance in SD4 (Sinninghe Damsté et al., 2011a, 2018). Small amounts of ether-bound *iso*-
diabolic acid and its derivatives, including brGDGT Ia, have been detected in two Acidobacteria SD1 species
isolated from soil. So far, only SD4 species have been shown to produce 5-methyl *iso*-diabolic acid derivatives,





whereas the other SDs formed 6-methyl *iso*-diabolic acids. This suggested that the position of methylation of *iso*-
diabolic acid may be controlled by phylogenetic affiliations within the Acidobacteria and thus may not be a
direct but indirect response to environmental conditions (Sinninghe Damsté et al., 2018). Only little is known
about the occurrence and diversity of Acidobacteria in lakes (e.g., Zimmermann et al., 2012; Parvenova et al.,
2016; Preheim et al., 2016), but the concentrations of some individual brGDGTs in Lake Lugano at one instance
during the seasonal cycle (in contrast to the prolonged monthly sampling realized here) showed a strong
empirical correlation ($R^2 >0.56$) with the abundance of certain Acidobacteria SDs (i.e. 3, 5, 6, 8, 15, 17, 21;
Weber et al., 2018).
In our data from Lake Chala, only few correlations between individual brGDGTs and Acidobacteria
SDs are at least moderately strong (Fig. 8A). Surprisingly, whereas in Lake Chala the concentration of brGDGTs
Ib and IIb correlates with the abundance of SD21 Acidobacteria, in Lake Lugano this same SD correlates with
the concentrations of brGDGTs IIa, IIIa and IIIa″ instead (Weber et al., 2018). The weaker correlation observed
in Lake Chala may be partly due to the method of analysis; in contrast to the studies of Weber et al. (2018) and
Sollai et al. (2019) we determined the sum of core and intact polar lipid-derived individual brGDGTs rather than
the intact polar lipids separately. Intact polar lipids are generally considered to be better markers for 'live'
bacteria because the polar head group is thought to be lost quickly after cell death (White et al., 1979; Harvey et
al., 1986). However, given the overall low abundance of Acidobacterial 16S rRNA sequences in Lake Chala
SPM (on average 0.2% of total prokaryotes, with values up to 6%), the omnipresence and high concentrations of
brGDGTs (Fig. 4), and the mostly weak correlation between them, it seems unlikely that Acidobacteria are the
predominant producers of brGDGTs in Lake Chala.
To investigate alternative bacterial sources for the brGDGTs, we also correlated their individual
concentrations with those of the bacterial taxa identified in the SPM (Fig. 8B). Although empirical co-occurrence
of brGDGTs and microbial taxa alone does not suffice to reveal the exact source organism(s) of those brGDGTs,
the detected phyla might either contain brGDGT-producing organisms or be associated with similar habitats. For
example, correlation of brGDGT Ib and IIb with the Actinobacteria phylum may be indicative of the depth
habitat or growth season of the organism producing these specific brGDGTs. Our broad brGDGT-16S rRNA
data comparison clusters different groups of brGDGTs, and can broadly be defined as Ib+IIb, non-cyclic 5-
methyl brGDGTs, and 6-methyl brGDGTs, which all relate to a different bacterial composition (Fig. 8B). The
clustering is consistent with the spatiotemporal alternation of brGDGTs Ib+IIb and IIa′+IIIa′ observed in the
water column of Lake Chala (Fig. 4), and suggests that these different brGDGTs (i.e., 5-methyl *vs* 6-methyl, and
cyclic *vs* non-cyclic brGDGTs) may be produced by different (groups of) bacteria. Despite our extensive SPM
dataset, it is at this stage not possible to determine exactly which aquatic bacteria produce the brGDGTs in Lake
Chala but the higher correlations with bacterial taxa other than Acidobacteria suggests that an exclusive origin of
lacustrine brGDGTs by Acidobacteria is deemed unlikely, in agreement with earlier, less extensive work (Weber
et al., 2018).

**4.4. Congruence between brGDGT distributions in SPM and settling particles**
It was previously noted that Lake Chala SPM is deprived of terrestrial biomarkers (van Bree et al., 2018),
whereas they do occur in the lake's bottom sediments (e.g., Sinninghe Damsté et al., 2011b). Since the brGDGTs
associated with SPM appear to be produced within in the water column, a possible contribution of soil-derived



brGDGTs to the lake sediments may be recognized in settling particles collected monthly in the sediment trap at
35 m, which represents a depth- and time-integrated signal of the upper water column, although it should be
realized that the position of the trap is located above the predominant zone of aquatic brGDGT production (Fig.
4). The 53-month sediment-trap record fully encompasses our 17-month SPM time series and thus enables direct
comparison between the datasets. Notably, the brGDGT distribution in sediment-trap material is highly similar
to that in the SPM, both during the overlapping time period and averaged over the 53-month interval, and
dissimilar from the brGDGT distribution in catchment soils (Figs. 5A and 6E). It thus appears that also the vast
majority of brGDGTs in settling particles has an aquatic origin, and consequently can be expected to show the
same temporal trends as the brGDGTs in SPM. Indeed, the observed alternation in the summed fractional
abundances of brGDGTs Ib+IIb versus IIa′+IIIa′ in SPM can also be recognized in the settling particles (Fig. 7D-
E). Onset of upper water-column stratification is marked by the relative increase of one of these two groups: the
fractional abundance of IIa′+IIIa′ increased sharply at the onset of stratification in late 2010, 2012 and 2014,
whereas those of Ib+IIb increased in late 2011 and 2013. The similar behavior of brGDGTs in SPM and settling
particles is also reflected in the PCA including both sample series (Fig. 6E), where PC1 separates 5- and 6-
methyl brGDGTs resulting from the temporal alternation of Ib+IIb and IIa′+IIIa′. Although it is not clear which
environmental variable controls the predominance of either group in any one year, this pattern supports our
suggestion that changing brGDGT distributions in Lake Chala primarily reflect distinct seasonal and inter-annual
variation in the composition of its aquatic microbial community rather than a physiological response in a
compositionally stable resident community.

Notwithstanding their apparently similar dynamics, the sediment-trap and SPM time series are not fully

equivalent. The brGDGTs in settling particles are supposedly produced in the upper 35 m of the water column,
and indeed mostly plot with the SPM$_{abovetrap}$ samples in the PCA (Fig. 6E). However, for the overlapping 17-
month period the time-integrated brGDGT distribution in sediment-trap material appears more similar to the
abundance-weighted, depth- and time-integrated brGDGT signal in SPM from the mostly anoxic water column
below the sediment trap (> 35 m) than to that of the more oxic upper water column (< 35 m; Fig. 5A-B). This
suggests that a substantial contribution of brGDGTs to the sediment trap may occur during periods when the
oxycline is situated well above 35 m. In addition, material from deeper water layers can also reach the sediment
trap during mixing episodes (reaching down to *ca* 40-45 m in 2014, and to as much as 60 m in other years;
Verschuren et al., 2009).

Temporal variation in brGDGT concentrations in the Lake Chala SPM appears to respond mainly to the

seasonal cycle of stratification and mixing (Fig. 4). This pattern is not reflected in the 4.5-year sediment-trap
record, where fluxes of both settling particles as a whole and brGDGTs do not show a clear, recurring annual
pattern (Fig. 7A-B). Substantial temporal variation in the total flux and distribution of brGDGTs in Lake Chala
has been reported previously, based on analysis of settling particles collected monthly between November 2006
and August 2010 (Sinninghe Damsté et al., 2009; Buckles et al., 2014). This contrasts with findings from
sediment-trap studies in north-temperate lakes (e.g. Loomis et al., 2014b; Miller et al., 2018), where brGDGT
distributions in settling particles remain relatively stable despite large seasonal variation in their fluxes. Aside
from the large temporal variation in brGDGT distributions in Lake Chala, the whole of our 17-month SPM
sampling period, and the period from September 2013 to September 2014 in particular, stand out in the 53-month
sediment-trap record due to the relatively high flux of 5-methyl brGDGTs with rings (Ib+IIb), and relatively low





flux of 6-methyl brGDGTs (IIa′+IIIa′) (Fig. 7D-E). The relatively large contribution of 5-methyl brGDGTs
during this period is reflected in low IR$_{6ME}$ values (< 0.7), otherwise uncommon in the entire 53-month time
series (Fig. 7C). Notably, this 13-month period of low IR$_{6ME}$ is also characterized by the near-absence of
terrestrial plant biomarkers in the SPM (van Bree et al., 2018), which may indicate that this distinct brGDGT
signature is representative of a primarily aquatic source of the brGDGTs. Indeed, any contribution of soil-
derived brGDGTs would be revealed by increased abundance of brGDGT-Ia, the dominant brGDGT in Chala
catchment soils (Buckles et al., 2014; Fig. 5A). However, the fractional abundance of brGDGT-Ia remains
relatively stable over the entire 53-month study period (Fig. 7F), implying that the vast majority of the brGDGTs
in the sediment-trap record have an aquatic origin. The caveat is that due to the mid-lake position of the sediment
trap and steeply sloping crater walls, we cannot exclude the possibility for soil material to be deposited on the
lake floor without reaching the sediment trap. This scenario may also explain the contrasting occurrence of
terrestrial biomarkers in surficial bottom sediments and the water column (i.e., SPM) of Lake Chala.
**4.5 Discrepancy between brGDGT signatures in the water-column, soils, and sediments**
The brGDGT distribution in bottom sediments of Lake Chala clearly differs from those in the SPM and settling
particles, even when distributions in the latter are integrated over time and weighted-averaged (Fig. 7C-F). For
example, the full range of IR$_{6ME}$ values is very wide in SPM (0.35-1.00) and settling particles (0.45-1.00)
whereas the three sediment samples have near-identical IR$_{6ME}$ values (~0.67) that are substantially lower than the
overall weighted-averaged IR$_{6ME}$ of the SPM and the settling particles (Fig. 7C). In other words, over 88% ($n$ =
17) or 96.2% ($n$ = 53) of brGDGTs II and III in SPM and settling particles belong to the 6-methyl variety (Fig.
7B), whereas this is only 67% in the bottom sediment. Also the brGDGT fractional abundances in the sediments
are different from those in the water column, and in particular those of the 5-methyl brGDGTs without
cyclopentane moieties (Ia, IIa, IIIa; Figs. 5 and 7). Interestingly, the sedimentary brGDGT signature is also
clearly different from that of the catchment soils (Figs. 5 and 7). Although this may be the result of mixed
aquatic and soil-derived brGDGTs, this cannot explain the fractional abundances of in particular brGDGT-IIa
and IIb in the sediment, which are higher than those in both the water column and the soils (Fig. 5A). These
differences in brGDGT signatures in the soils, water column and the sediment suggest that additional brGDGT
production may take place within the bottom sediments, as suggested previously (Buckles et al., 2014). Hence,
the final brGDGT signal that is stored in Lake Chala sediments is influenced by i) seasonal changes and
substantial inter-annual variability in aquatic brGDGT production in the water column, ii) production within the
sediments, and iii) varying proportions of aquatic and terrestrial brGDGTs over time, although the evidence for a
soil contribution to Lake Chala sedimentary GDGTs remains weak.
**4.6.    Implications for brGDGT-based paleoclimate reconstruction**
In order to use brGDGTs extracted from lake sediments for paleoclimate reconstruction, we need to understand
how the environmental parameters of interest (primarily temperature and pH) are reflected by the signal that is
finally exported to and preserved in the sedimentary record. An earlier study of brGDGTs in time series of
settling particles from Lake Chala (Buckles et al., 2014) indicated that mean annual air temperature (MAAT)
was underestimated by ~11-13 °C (estimates were 14±5 °C and 13±6 ºC, using the East African Lake (EAL)
calibrations of Tierney et al. (2010) and Loomis et al. (2012), respectively). Furthermore, maxima and minima in





reconstructed temperatures from the time series of settling particles lagged changes in air temperature by up to 5-
6 months. Buckles et al. (2014) attributed these offsets to a shifted ratio of aquatic versus soil-derived brGDGTs,
and also noted that the brGDGTs in Lake Chala may have a different relationship with temperature than those in
the EAL calibrations. Moreover, the co-elution of 5- and 6-methyl brGDGTs in their analysis may have
contributed to the observed offset. Application of the improved chromatography method (Hopmans et al., 2016)
in the current study allows us to use the most recent temperature calibration based on stepwise forward selection
of 5- and 6-methyl brGDGTs in the EAL dataset ($MAAT_{SFS}$; Russell et al., 2018). Application of this transfer
function to our sediment-trap record generates MAAT estimates between 18.5 and 25.2 ºC (on average 22.1±1.7
ºC), and a flux-weighted overall average of 22.8 ºC (Fig. 7G). Underestimation of the observed local MAAT
(25.1 ºC) is thereby reduced to about 2 ºC, i.e. in the range of the calibration error of 2.1 ºC.

Nevertheless, seasonal variation in reconstructed temperature based on settling particles does not seem

very consistent (Fig. 7G). Whereas highest air (and surface-water) temperature occurs during the period of strong
water-column stratification, brGDGT-based temperature inferences peak during the periods of stratification in
2010-2011, 2013-2014 and at the end of 2014, but also during the periods of deep mixing in 2011 and 2012 (Fig.
7G). As the temporal variation in brGDGT distributions in the water column of Lake Chala appears to be linked
to microbial community changes that are at best only indirectly related to temperate, this may be one reason why
brGDGT signatures do not clearly track measured MMAT (Fig. 7G). On the other hand, the modestly higher
temperature of the epilimnion compared to that of the lower water column (up to ~4 °C, depending on the
season) is reflected in a higher average reconstructed temperature for $SPM_{abovetrap}$ (23.4 ºC, $n = 72$) than
$SPM_{belowtrap}$ (21.5 ºC, $n = 71$, $p < 0.01$; Fig. 7G). Similarly, the decrease of lake-water pH with depth (Wolff et
al., 2014; van Bree et al., 2018) is reflected by higher reconstructed pH for the lake surface water (on average 8.5
at 0 m) than that of deeper water layers (on average 8.1 at 80 m; $p < 0.01$). Also the elevated surface-water pH
that may be expected to occur during peak primary productivity in the mixing season appears to be recorded by
brGDGTs in settling particles (8.1 during mixing ($n = 16$) versus 7.7 during stratification ($n = 37$; $p < 0.02$), even
though the 53-month time series of brGDGT-inferred pH does not follow clear seasonal trends (Fig. 7H).
Especially during the interval from September 2013 to September 2014 that is characterized by lower $IR_{6ME}$
values, the inferred pH is nearly constant (as is, incidentally, the observed pH: Fig. 7H). This low $IR_{6ME}$ is the
net result of the relative increase in cyclopentane moieties (Ib+IIb, Fig. 7F) and decrease in the degree of
isomerization (more 5-methyl brGDGTs, Fig. 7C), whereas they are both positively related to pH at least in
global soils (Weijers et al., 2007; De Jonge et al., 2014). The opposite trends in the degrees of cyclisation and
isomerization of brGDGTs in settling particles of Lake Chala also may explain the generally weak relationship
between bottom-sediment brGDGT distribution and surface-water pH in the EAL dataset (Tierney et al., 2010;
Loomis et al., 2014; Russell et al., 2018), and supports the suggestion made by these authors that another
variable than pH is responsible for changes in brGDGT signatures in lakes.

Notably, average MAAT and pH values inferred from the brGDGTs in surficial bottom sediments are

yet again different from those based on brGDGTs produced in the water column and in catchment soils (Fig. 7G,
H). The distinct signature of the sedimentary brGDGTs (Fig. 5A) suggests that besides an aquatic source and a
potential, but unlikely, soil contribution, those brGDGTs are partly produced within the sediment. Regardless,
sediment-inferred MAAT (21.9 ºC) and pH (9.1) generated with the most recent EAL calibrations (Russell et al.,
2018) are within reasonable range of measured MAAT (25.1 ºC) and surface water pH (9.0), consistent with the





idea that the brGDGTs in lake sediments carry (albeit indirectly) truthful environmental information (e.g.,
Tierney et al., 2010; Russell et al., 2018), at least in the modern system. It suggests that the brGDGT-
temperature calibrations already take the *in-situ* sedimentary production into account. If this principle holds over
longer timescales, it implies that we can use brGDGTs in lake sediments for paleoclimate reconstructions, even
without fully understanding the mechanism that determines their signature.

**5.   Conclusions**
BrGDGTs in the water column of Lake Chala are primarily produced *in situ*. The amounts and distributions of
individual aquatic brGDGT compounds are highly variable with depth and over time, and do not consistently
relate to ambient temperature, pH or oxygen but still appear to respond to the seasonal alternation of water-
column mixing and stratification. The aquatic brGDGT assemblage is alternatively dominated by the compounds
Ib+IIb and IIa′+IIIa′, with each pair linked to the occurrence of different bacterial taxa, other than, or besides the
Acidobacteria. Hence, temporal changes in brGDGT assemblages are likely due to the sequential occurrence of
different groups of aquatic bacteria producing different types of brGDGTs, rather than by membrane adaptation
within one group. BrGDGTs in settling particles reveal substantial inter-annual variation in the bacterial
community of this tropical lake, superimposed on seasonal variation. Although the brGDGT distributions in
SPM and settling particles from Lake Chala cannot be directly linked to local variation in air or water
temperature, temporally-integrated and flux-weighted brGDGT compositions do produce reasonable temperature
and surface-water pH estimates when using the new EAL calibration of Russell et al. (2018). Regardless, the
distinct brGDGT signature of surficial bottom sediments suggests that part of the sedimentary brGDGT pool is
produced within the sediment itself. It thus remains crucial to discover the producers of brGDGTs, and the
general drivers of brGDGT production, in lakes so that the uncertainties in lacustrine paleothermometry can be
further constrained.

**Acknowledgements**
We thank C.M. Oluseno for conducting the monthly lake sampling and monitoring. We thank A. Negash and P.
de Regt for lipid extractions, and J.W. de Leeuw for feedback on the manuscript. We are grateful to A. van Dijk,
D. Kasjaniuk, A. van Leeuwen-Tolboom and K. Nierop at Utrecht University; and M. Baas, D. Dorhout, E.C.
Hopmans, A. Mets, J. Ossebaar, S. Vreugdenhil and M. Brouwer at the Royal NIOZ for technical and analytical
support. We furthermore thank A. Roepert for help with R, and C. De Jonge for discussions on brGDGTs.
Fieldwork with collection of the studied sample materials was carried out with permission from the government
of Kenya through permit 13/001/11C to D.V. In accordance with National Environmental Management
Authority regulations in the context of the Nagoya Protocol, DNA extracts of the analyzed suspended-particulate
samples are archived at the National Museums of Kenya (NMK), under voucher numbers NMK:BCT:80001 to
NMK:BCT:80221; we thank A. Mwaura and S.M. Rucina for facilitation. The raw data of the 16S rRNA gene
amplicon reads were deposited in the NCBI Sequence Read Archive (SRA); BioProject number upon request.
This research was supported by the NESSC Gravitation Grant (024.002.001) from the Dutch Ministry of



Education, Culture and Science (OCW) and the European Research Council (ERC) under the European Union's
Horizon 2020 research and innovation program (grant agreement no. 694569 – MICROLIPIDS) both to J.S.S.D.



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





**Figure captions**

Figure 1. Molecular structures of brGDGTs, consisting of two ether-linked dialkyl chains with zero to two additional methyl branches (I, II and III) and zero to two cyclopentyl moieties (suffixes a, b and c). The 6-methyl isomers are denoted with a prime. Compounds indicated in bold are the eight most common brGDGTs encountered in Lake Chala, and focused upon in data presentation and discussion.

Figure 2. Bathymetry of Lake Chala situated within its steep-sided crater catchment (outer bold full line), with sampling locations of suspended particulate matter (SPM; black square), settling particles (sediment trap; open triangle), surficial lake-bottom sediments (grey triangles), and terrestrial soils (open circles). Bathymetry adapted from Moernaut et al. (2010).

Figure 3. Temperature (ºC) variation within the water column of Lake Chala between September 2010 and January 2015, based on automatic loggers suspended at 2, 10, 20, 25, 50 and 80 m depth (as available), in relation to mean monthly air temperature (MMAT; stippled line, Buckles et al., 2014). The dark blue line shows the position of the oxycline between September 2013 and December 2015, based on the shallowest depth with dissolved oxygen concentration <0.2 mg L$^{-1}$ as measured by monthly water-column profiling (van Bree et al., 2018). Grey shading highlights the seasonal periods of upper

water-column stratification (S) and deep mixing (DM). Due to a hiatus in temperature logging data, timing of the start and end of the deep-mixing period in 2012 was inferred from the MMAT trend.

Figure 4. Depth-interpolated concentrations (in ng L$^{-1}$) of the summed and eight most common individual brGDGT compounds in SPM collected at eight depth intervals (occasionally 13) between 0 and 80 m in Lake Chala, at approximately

955 monthly intervals between September 2013 and January 2015. Also indicated is the varying position of the oxycline (bold stippled line) in relation to the static position of the sediment trap at 35 m depth (thin dashed line), which separates the SPM$_{abovetrap}$ and SPM$_{belowtrap}$ zones. Grey background shading indicates the seasonal periods of upper water-column stratification (S) and deep mixing (DM), as in Fig.3.

Figure 5. Average distribution of brGDGTs in the various sets of samples analyzed in this study. A: Temporally-integrated,

concentration- or flux-weighted average fractional abundances of individual brGDGT compounds in SPM from above (light blue) and below (dark blue) the sediment trap, and settling particles trapped over the 17-month period of SPM sampling (Sept-2013 to Jan-2015; light green) and over the longer 53-month period starting three years earlier (Sept-2010 to Jan-2015, dark green), compared with average fractional abundances of the same brGDGTs in surficial lake-bottom sediments (orange) and catchment soils (red). B: Proportion of tetra-, penta- and hexamethyl brGDGTs in SPM from above (light blue circles)

and below (dark blue circles) the sediment trap, and in settling particles (green squares), lake sediments (orange diamonds)



and soils (red triangles) plotted over corresponding data from the surficial bottom sediments of 65 East African Lakes (grey triangles; Russell et al., 2018).

Figure 6. Principal component analysis (PCA) of the fractional abundances of the eight major brGDGTs in SPM ($n$ = 143) and settling particles ($n$ = 53) from Lake Chala. A-B: PC1 vs PC2 (A) and PC2 vs PC3 (B) of the SPM samples, with black vectors indicating the PCA scores of individual brGDGTs, and blue vectors showing the PCA scores of environmental variables added passively. Temperature and pH are measured (0-50 m depth, van Bree et al., 2018) and assumed constant from 50 m to 80 m depth. C-D: PC1 vs PC2 (C) and PC2 vs PC3 (D) of the settling particles, with black vectors indicating the PCA scores of individual brGDGTs, and blue vector showing the PCA score of the total bulk settling flux added passively. E: Combined PCA of the fractional abundances of the (mainly aquatic) brGDGTs in all SPM (blue circles) and settling-particle (green squares) samples, with distinction between SPM from above (light blue) and below (dark blue) the sediment trap. The PCA scores of lake sediments (orange diamonds) and soils (red triangles) were added passively.

Figure 7. Time series of settling-particle data from Lake Chala, based on 53 months of sediment-trap deployment between August 2010 and January 2015. A: Temporal variation in total bulk dry flux (mg m$^{-2}$ day$^{-1}$). B: Total brGDGT flux (ng m$^{-2}$ day$^{-1}$) with indication of the proportions of tetra-methylated (green), 5-methyl (purple) and 6-methyl (brown) brGDGTs. C: Fraction of 6-methyl penta- and hexamethylated brGDGTs (IR$_{6ME}$). D-F: Fractional abundances of brGDGTs Ib+IIb (D), IIa′+IIIa′ (E) and Ia (F). G: Reconstructed mean annual air temperature (MAAT), using the EAL calibration of Russell et al. (2018). Also indicated are MMAT (red dashed line) and MAAT (black dashed line). H: Reconstructed surface-water pH, using the EAL calibration of Russell et al. (2018), and pH measured at the surface (0 m) during the period September 2013 to January 2015 (van Bree et al., 2018). The right-hand panels show boxplots indicating median, interquartile, minimum, maximum and outlier values of the bulk and brGDGT fluxes and proxies (A-B), suspended-particulate data (SPM in C-H; $n$ = 143), settling-particle data (ST in C-H; $n$ = 53), lake-sediment data (SED in C-H; $n$ = 3) and catchment soil data (SOIL in C-H; $n$ = 7). These box plots are superimposed with flux- or abundance-weighted average values of the same for SPM$_{abovetrap}$ (light blue circle), SPM$_{belowtrap}$ (dark blue circle), settling particles trapped over the 17-month period of SPM sampling (September 2013 to January 2015, crossed green square) or over the 53-month period starting three years earlier (September 2010 to January 2015, green square), lake sediments (orange diamond) and soils (red triangle). Grey background shading highlights the seasonal periods of upper water-column stratification (S) and deep mixing (DM).

Figure 8. Correlation matrix ($R^2$; represented by shades of blue) between the absolute concentration of the eight major brGDGTs in Lake Chala SPM (ng L$^{-1}$) and estimated 16S rRNA gene abundances (copies L$^{-1}$, see details in text). A: Acidobacteria SD 6, 18, 19 and 21, and the sum of all Acidobacteria. The Acidobacterial SD OTUs are present in at least 10% of the SPM samples measured for both brGDGTs and gene abundances. Only SDs that correlate with at least one brGDGT with an $R^2 \geq 0.05$ are shown. B: The 14 taxa of bacteria displaying highest correlation with individual brGDGTs,





divided in clusters of highest correlation with Ib and IIb or with the other brGDGTs. The bacterial OTUs are present in at

least 10% of the SPM samples measured for both biomarkers and gene abundances. Only SDs that correlate with at least one

brGDGTs with $R^2 \geq 0.2$ are shown.





**FIGURE 1**

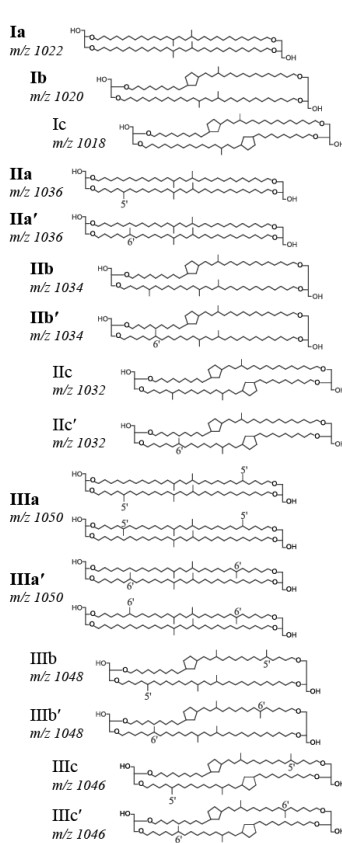





FIGURE 2

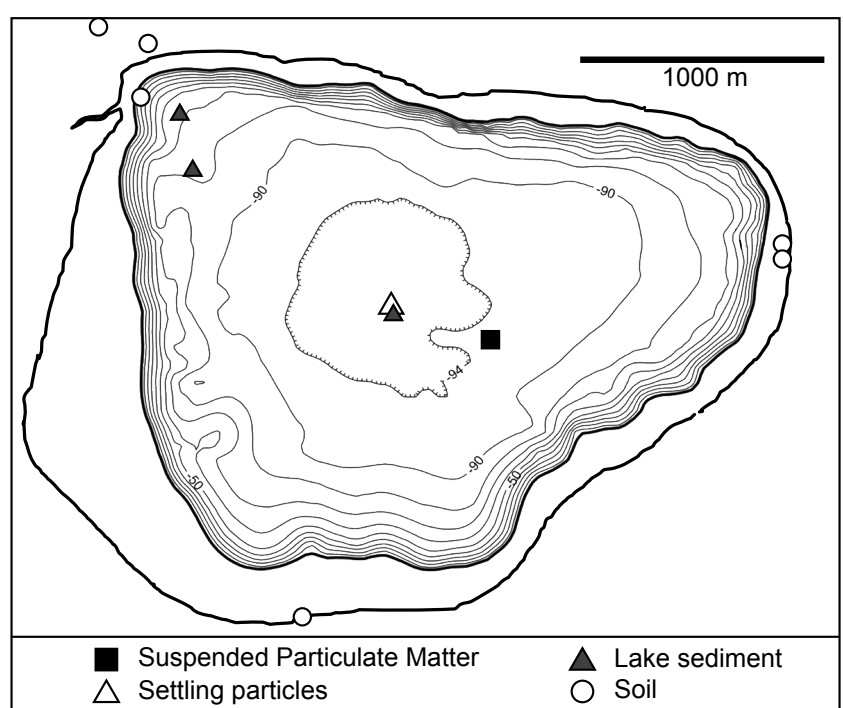



FIGURE 3

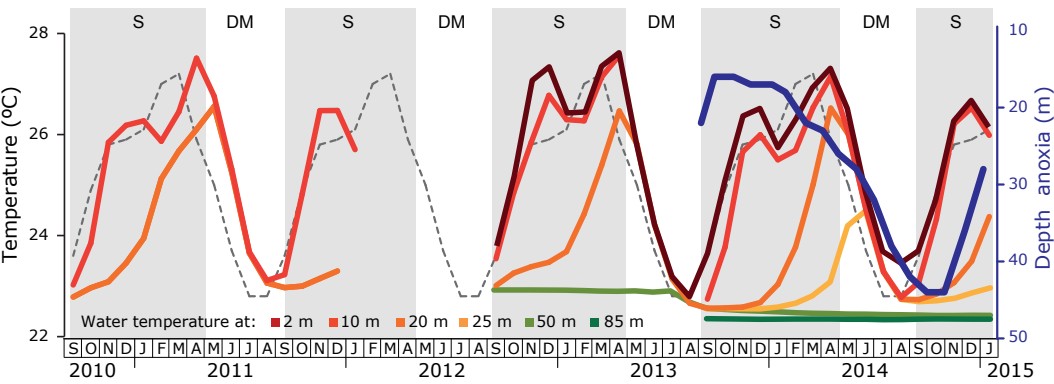





FIGURE 4

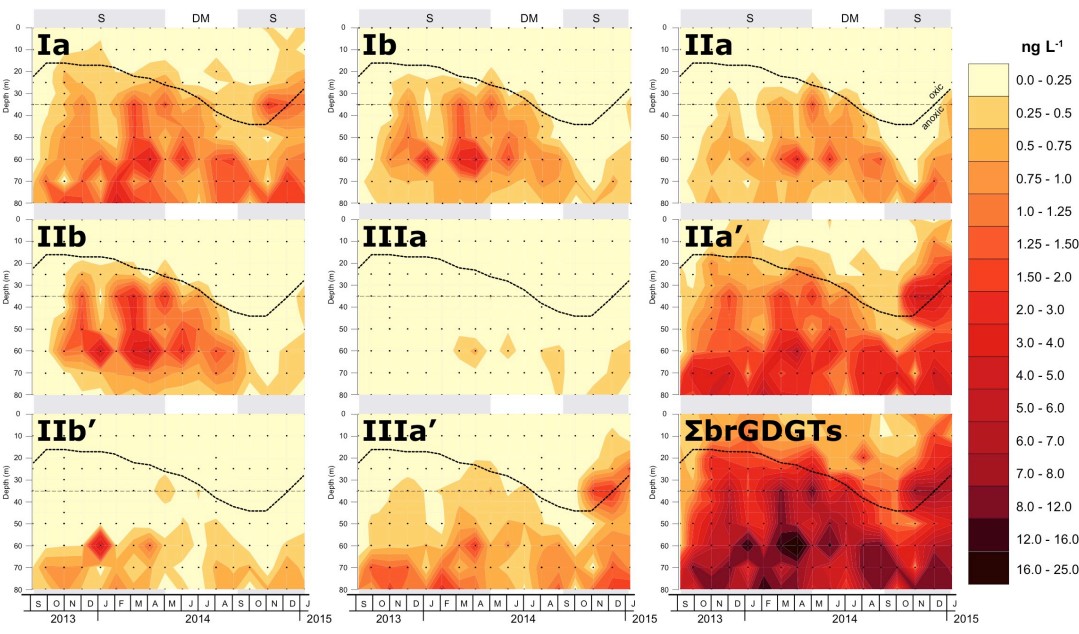



FIGURE 5

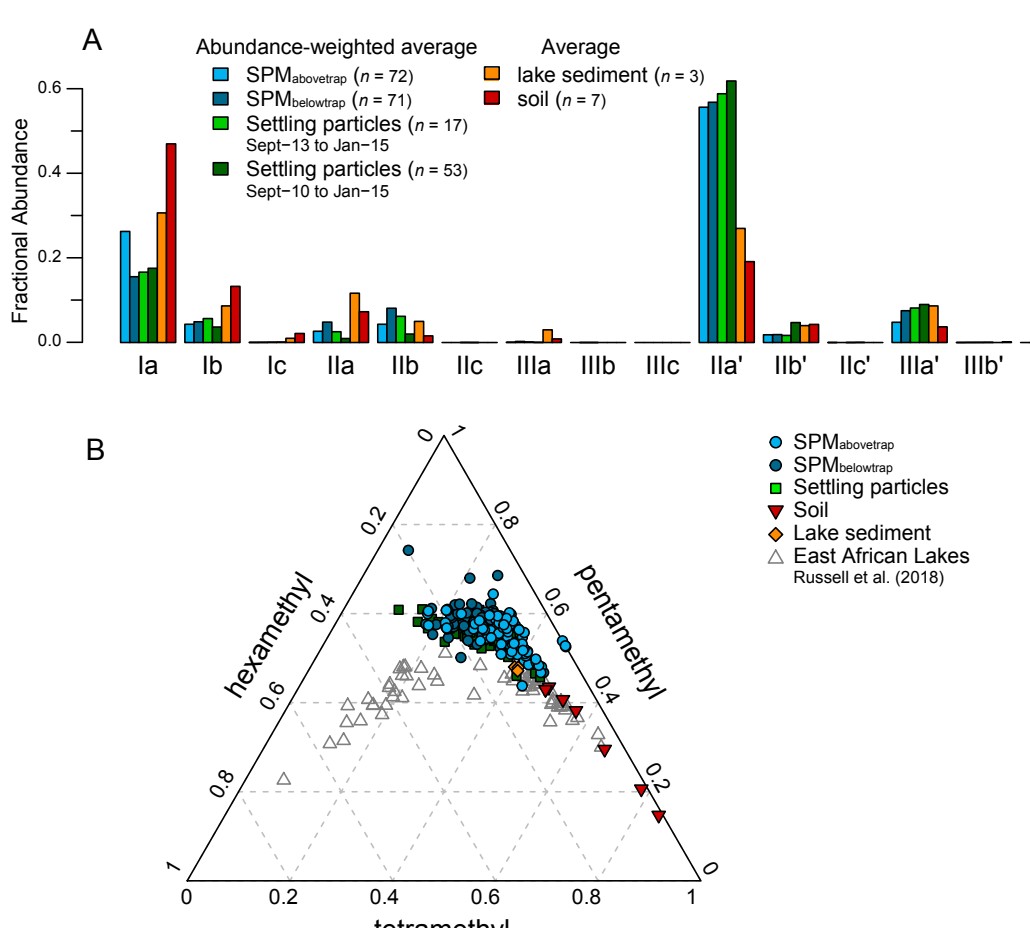



FIGURE 6

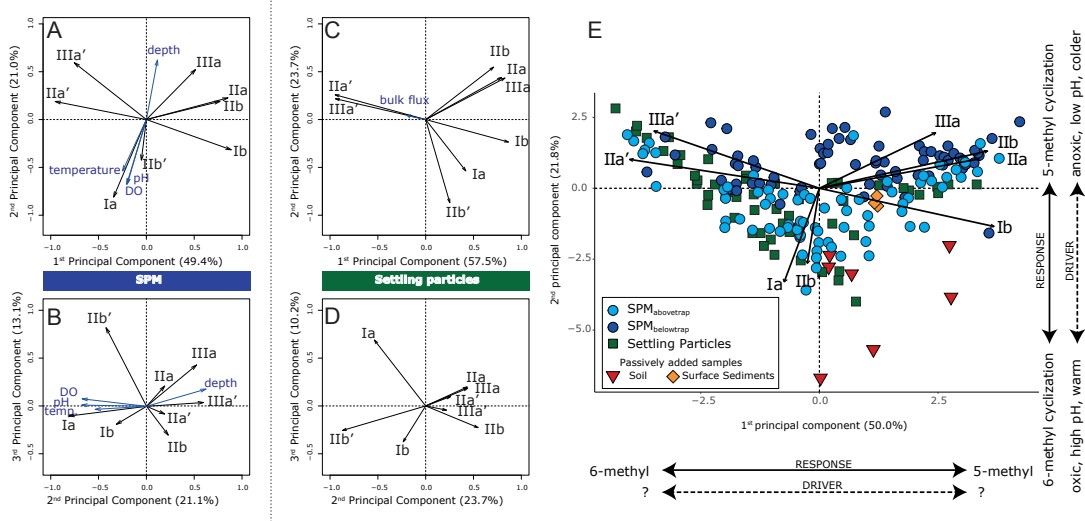





FIGURE 7

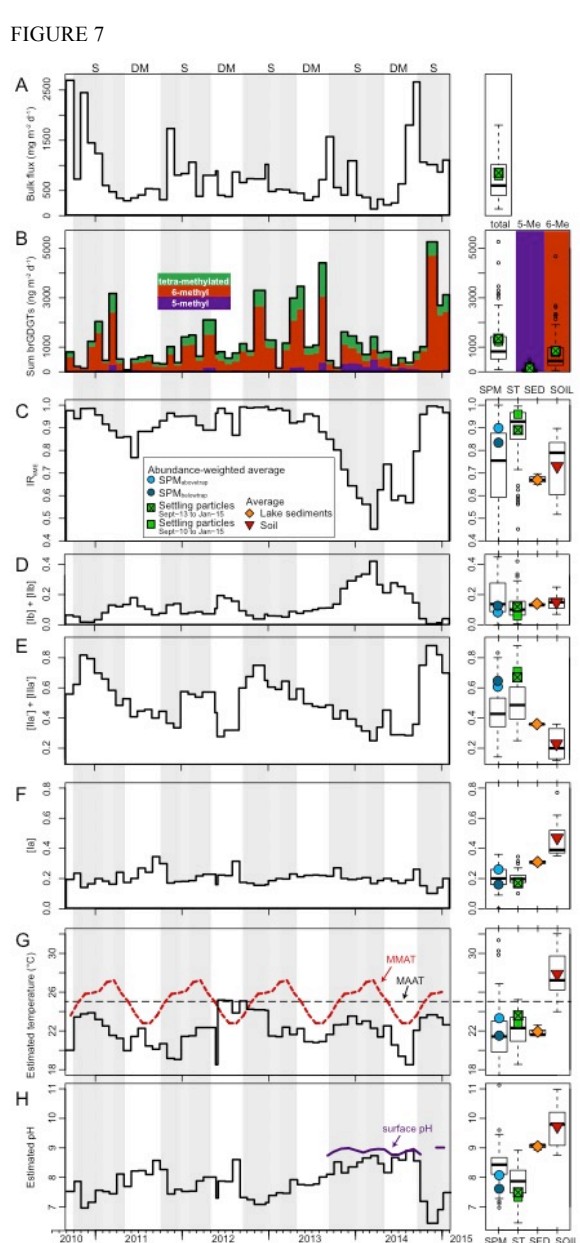





FIGURE 8

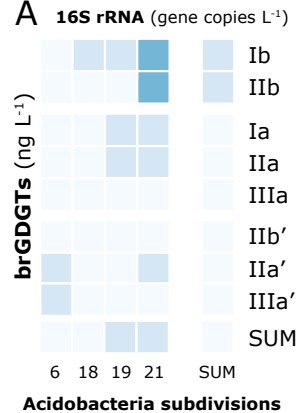

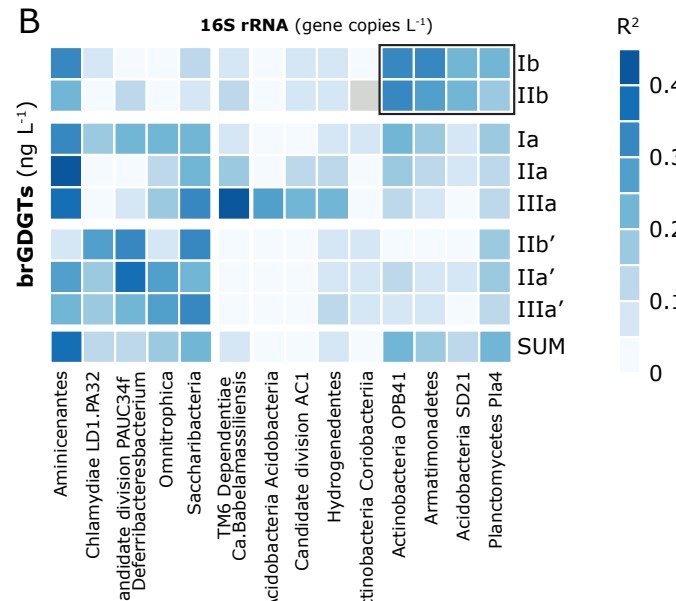
