# Peer review of "Seasonal variability and sources of *in situ* brGDGT production in a permanently stratified African crater lake"

_Biogeosciences, 2020_

## Referee Comment (RC1) · Anonymous Referee #1 · 25 Aug 2020

This is a well written and timely contribution, focusing on the distribution of branched GDGTs in Lake Chala, Kenya. The data support the conclusions, and move forward our understanding these biomarkers as potential environmental proxies.

Main points: 1. brGDGTs produced primarily in situ in the water column of Lake Chala a. distinct distribution from soils b. also appears to be production in sediments, as they are distinct from water and soil brGDGTs 2. brGDGT distribution changes with depth and season, but not exactly as expected if driven by environmental parameters 3. Likely this is due to production by various different bacterial groups producing at different times, rather than a single group changing lipids 4. brGDGTs do not correlate

well with Acidobacteria rRNA, suggesting other groups are more dominant producers

The research discussed here is an important step towards determining source organisms of brGDGTs and the extent to which their distribution actually reflect temperature, pH, or other parameters.

I'd like to see some discussion of how this system is unique, and whether they think that would bias the results. For example, Chala is a crater lake with very little terrestrial input. What about a system with very different morphology that does get a lot of terrestrial input? This sort of reflection would be easy to add to the implications section, and would figure strongly in how broadly the results from Chala are likely to apply.

I did notice a few typos scattered throughout (e.g. line 199 should read "…liquid chromatograph…"; line 648 should read "…related to temperature…"), but one more detailed read by the authors should find these.

---

## Referee Comment (RC2) · Anonymous Referee #2 · 10 Sep 2020

General Comments

The paper by van Bree and coauthors lays out a clever study to examine crucial questions on the relationship between measured brGDGTs and climate parameters. Their main takeaway is that measured brGDGTs likely represent variations in bacterial communities that respond to seasonal stratification change, and thus downcore are indirectly, but significantly, related to climate parameters such as temperature. They thoroughly examine this in a very clear way by taking time series data from all potential sources of brGDGTs, along with DNA data, and then use rigorous statistical comparisons to demonstrate their findings. I think the writing and organization is very clear,

logical, and direct, and I was left without a doubt that their findings are backed with robust analyses. I learned a lot reading this manuscript and I recommend it for publication. Please find my specific comments/questions along with technical corrections and comments on the paper's figures below.

Specific Comments

Lines 37-38: What does (sub-) mean? Are you talking about the tropics or subtropics? Speleothems are in both?

Lines 46-47: Why is temperature the most important climate parameter to reconstruct? Particularly if you're focusing on the tropics, temperature doesn't change much.

Really nice thorough introduction, very clear and informative

Please include a brief (one sentence) analysis of the samples run in the two different labs. Was there any significant different in any way? If not, I think just stating that there were no discernable differences would be fine.

Be clearer at the end of the discussion section about the drawbacks of applying brGDGTs to downcore studies. What time scales would work best? Could we trust the absolute T values, or focus on variability?

I think the authors should consider adding some of the folks named in the acknowledgments to the author list if they contributed a large amount of work, particularly the first three people, which is sounds like they did.

Technical Corrections

Line 16: SPM is defined, but not used in the rest of the abstract

Line 48: revise 'supposed to be' to 'supposedly'

Line 61: remove comma

Line 63: remove comma after 'additional'

Line 67: remove comma after '(IIIa")'

Line 68-71: this sentence is unclear due to the structure and perhaps misplacement of the word 'for' and 'with'. Please revise.

Line 75: add comma after 'Also', but I suggest using a different word, such as 'Further'.

Line 81: add comma after 'soils'

Line 83: add comma after 'factors'

Line 84: separate out light and mixing regime even if they're from the same study.

Line 85: add comma after 'chemistry'

Line 86: add comma after '2010)'

Line 103: SPM is defined as suspended particles, but perhaps use suspended particulate matter

Line 190: revise to 'into apolar, neutral and polar fractions'

Line 192: remove 'then' and revise 'similar' to 'similarly'

Line 193: rewrite this sentence perhaps to something like 'The lake sediment samples were also extracted and processed like the SPM'.

Line 203: what is Ø?

Line 423: remove 'is shown in Fig. 8' and reword so that Fig. 8 is just cited

Lines 427 and 519: formatting of R2 >0.2, the spacing is weird

Line 454-456: rephrase or remove the 'on the one hand' and 'on the other'

Line 491: add comma after 'Chala'

Line 515: remove 'but indirect' or add commas around it

Lines 521-522: rephrase to be more direct. 'only a few of the comparisons between. . ... are moderately correlated' or something.

Line 546: change 'less extensive' to something less negative

Line 555: change 'realized' to 'noted'

Line 559: remove 'also'

Line 648: change 'temperate' to 'temperature'

Line 666: add 'other' after 'variable'

Line 683: add something like ', which is indirectly related to climate' at the end of the sentence.

Formatting inconsistencies

Include n values with your r2 and p values, particularly when you make a conclusion from the relationship (or lack there of)

Sometimes the Oxford comma is used (e.g. title of section 2.2.1.), but often it's not. Either is fine, but be consistent.

Sometimes supplemental info is cited in the text as 'Table S.2' and sometimes as 'Table S4'. Either is fine, but be consistent.

Sometimes °C follows the temperature value directly, and other times there's a space between. Either is fine (I think?), but be consistent.

Sometimes R2=# and sometimes R2 = #. Either is fine, but be consistent.

Figures

In general, they look nice, but appear as different people made different plots – I suggest using consistent fonts throughout. Also, the fonts are very, very small. I view the plots as full page graphs, and can still barely see a lot of the axis labels and numbers,

particularly in figures 4, 6, and 7. This is crucial to change.

Fig. 2: Show where in East Africa this is with an inset map.

Fig. 3: The key is confusing because not all the colors are in it. I understand that the descriptions are in the caption, but perhaps have a clearer key with differently weighted lines. For instance, for the water T at various depths have solid lines in different colors (perhaps a bit thinner than what's there), as you do, and then different dashed lines for other variables plotted.

Fig. 4: Font is very small... will need to be whole page to barely see it. Perhaps get ride of the y axis labels on the second and third columns, smush them together, and make the axes and colorbar font much larger

Fig. 7: very blurry, but perhaps that's an artifact of the submission system. If not, it'd be great to update it to higher resolution.

Fig. 8: looks really nice!

---

## Author Comment (AC1) · 14 Sep 2020

The comment was uploaded in the form of a supplement:
https://bg.copernicus.org/preprints/bg-2020-233/bg-2020-233-AC1-supplement.pdf

---

## Author Comment (AC2) · 14 Sep 2020

**Author Reply to Anonymous Referee #2**

General Comments:
The paper by van Bree and coauthors lays out a clever study to examine crucial questions on the relationship between measured brGDGTs and climate parameters. Their main takeaway is that measured brGDGTs likely represent variations in bacterial com-munities that respond to seasonal stratification change, and thus downcore are indirectly, but significantly, related to climate parameters such as temperature. They thoroughly examine this in a very clear way by taking time series data from all potential sources of brGDGTs, along with DNA data, and then use rigorous statistical comparisons to demonstrate their findings. I think the writing and organization is very clear, logical, and direct, and I was left without a doubt that their findings are backed with robust analyses. I learned a lot reading this manuscript and I recommend it for publication. Please find my specific comments/questions along with technical corrections and comments on the paper's figures below.

*Reply: We thank the referee for this positive feedback on our manuscript.*

Specific Comments:
Lines 37-38: What does (sub-) mean? Are you talking about the tropics or subtropics? Speleothems are in both?

*Reply: We refer to both tropics and subtropics in our text as an area where lake sediment may provide valuable archives of continental climate history, as opposed to ice cores. Speleothems are indeed also used for tropical paleoreconstructions, so we will no longer mention them in the revised version.*

Lines 46-47: Why is temperature the most important climate parameter to reconstruct? Particularly if you're focusing on the tropics, temperature doesn't change much.

*Reply: While temperature change in the tropics is indeed relatively modest even on glacial-interglacial time scales (3-4 °C at sea level; e.g. Loomis et al., 2017; Chevalier et al., 2020), this has major impact on tropical continental rainfall through its control on sea-surface evaporation and on the temperature contrast between the ocean and adjacent continents. Therefore, no rainfall or moisture-balance reconstruction from the tropics can be properly interpreted without knowing local/regional temperature history as reference frame. More broadly, a record of past tropical temperature evolution is needed as low-latitude end-member to determine the meridional temperature gradient, and to reconstruct global heat distribution, through time.*

Really nice thorough introduction, very clear and informative. Please include a brief (one sentence) analysis of the samples run in the two different labs. Was there any significant difference in any way? If not, I think just stating that there were no discernable differences would be fine.

*Reply: The instruments in both labs are tuned towards the same standards. Although the sensitivity (and thus detection limit) of the mass spectrometer was slightly different between labs, this has had no influence on the brGDGT-based indices and proxy values. We will add a sentence on this topic in the revised manuscript.*

Be clearer at the end of the discussion section about the drawbacks of applying brGDGTs to downcore studies. What time scales would work best? Could we trust the absolute T values, or focus on variability?

*Reply: As can be seen in our 53-month record of brGDGTs in settling particles (Fig. 7), there is no clear recurrent pattern in both the flux and composition of brGDGTs, and a direct link to temperature is also absent. Interestingly, the flux-weighed average brGDGT signal in the settling particles, as well as that of the sediments, translate into a temperature that is close to the measured temperature using the Russell et al. (2018) East African Lake calibration. This indicates that brGDGTs can be used to reconstruct absolute temperatures for sediments that integrate at least several years. We will clarify this at the end of the discussion section in our revised manuscript.*

I think the authors should consider adding some of the folks named in the acknowledgments to the author list if they contributed a large amount of work, particularly the first three people, which sounds like they did.

*Reply: We thank the referee for the suggestion. However, the contributions to this study made by the people named in the acknowledgements are of a technical (supportive) rather than scientific (interpretative) nature. We feel confident in having properly defined as author those people who made a scientific contribution to this work.*

Technical Corrections:

*Reply: Thank you for pointing out these (mostly) textual issues. We will address them during the revisions. As also indicated in our reply to reviewer #1, we will carefully re-read our manuscript and make further textual corrections where appropriate.*

Line 16: SPM is defined, but not used in the rest of the abstract
Line 48: revise 'supposed to be' to 'supposedly'
Line 61: remove comma
Line 63: remove comma after 'additional'
Line 67: remove comma after '(IIIa")'
Line 68-71: this sentence is unclear due to the structure and perhaps misplacement of the word 'for' and 'with'. Please revise.

*Reply: we will revise this sentence as follows: "Furthermore, comparison of the stable carbon-isotopic composition ($\delta^{13}C$) of brGDGTs from lakes and nearby soils indicates distinctive signatures, and thus sources of the lacustrine and soil-derived brGDGTs, with lacustrine brGDGTs being significantly more $^{13}C$-depleted (Weber et al., 2015; 2018; Colcord et al., 2017)".*

Line 75: add comma after 'Also', but I suggest using a different word, such as 'Further'.
Line 81: add comma after 'soils'
Line 83: add comma after 'factors'
Line 84: separate out light and mixing regime even if they're from the same study.
Line 85: add comma after 'chemistry'
Line 86: add comma after '2010)'
Line 103: SPM is defined as suspended particles, but perhaps use suspended particulate matter

Line 190: revise to 'into apolar, neutral and polar fractions' Line 192: remove 'then' and revise 'similar' to 'similarly'

Line 193: rewrite this sentence perhaps to something like 'The lake sediment samples were also extracted and processed like the SPM'.

Line 203: what is Ø?

*Reply: Ø refers to the diameter of the BEH particles in the column.*

Line 423: remove 'is shown in Fig. 8' and reword so that Fig. 8 is just cited

Lines 427 and 519: formatting of R2 >0.2, the spacing is weird

Line 454-456: rephrase or remove the 'on the one hand' and 'on the other'

Line 491: add comma after 'Chala'

Line 515: remove 'but indirect' or add commas around it

Lines 521-522: rephrase to be more direct. 'only a few of the comparisons between. . ... are moderately correlated' or something.

Line 546: change 'less extensive' to something less negative

Line 555: change 'realized' to 'noted'

Line 559: remove 'also'

Line 648: change 'temperate' to 'temperature'

Line 666: add 'other' after 'variable'

Line 683: add something like ', which is indirectly related to climate' at the end of the sentence.

Formatting inconsistencies:

Include n values with your r2 and p values, particularly when you make a conclusion from the relationship (or lack there of).

Sometimes the Oxford comma is used (e.g. title of section 2.2.1.), but often it's not. Either is fine, but be consistent.

Sometimes supplemental info is cited in the text as 'Table S.2' and sometimes as 'Table S4'. Either is fine, but be consistent.

Sometimes $^{\circ}$C follows the temperature value directly, and other times there's a space between. Either is fine (I think?), but be consistent.

Sometimes R2=# and sometimes R2 = #. Either is fine, but be consistent.

*Reply: we will check our manuscript for these inconsistencies and make changes accordingly.*

Figures:

In general, they look nice, but appear as different people made different plots – I suggest using consistent fonts throughout. Also, the fonts are very, very small. I view the plots as full page graphs, and can still barely see a lot of the axis labels and numbers, particularly in figures 4, 6, and 7. This is crucial to change.

Fig. 2: Show where in East Africa this is with an inset map.

Fig. 3: The key is confusing because not all the colors are in it. I understand that the descriptions are in the caption, but perhaps have a clearer key with differently weighted lines. For instance, for the water T at various depths have solid lines in different colors (perhaps a bit thinner than what's there), as you do, and then different dashed lines for other variables plotted.

Fig. 4: Font is very small. . . will need to be whole page to barely see it. Perhaps get ride of the y axis labels on the second and third columns, smush them together, and make the axes and colorbar font much larger

Fig. 7: very blurry, but perhaps that's an artifact of the submission system. If not, it'd be great to update it to higher resolution.
Fig. 8: looks really nice!

*Reply: we thank the referee for the feedback on the figures. We will carefully consider the suggested changes and revise the figures accordingly.*

References:
Chevalier et al., 2020 Geology, in press (https://doi.org/10.1130/G47841.1)
Loomis et al., 2017 Science Advances 3, e1600815
Russell et al., 2018 Org. Geochem 117, 56-69

---

## Author Response (AR1)

**Point to point reply to referee comments**

Dear editor,

We have revised our manuscript 'Seasonal variability and sources of in situ brGDGT production in a permanently stratified African crater lake' based on the comments of two anonymous referees. Please find a list of our point-to-point changes below.

Besides the comments of these referees, we have replaced the extrapolated measured temperature data from Buckles et al. (2014) in Figs. 3 and 7 with measured temperature data from Bodé et al. (2020) to match the time interval of the sediment trap time series. Note that this replacement has no influence on the interpretation of our data or the conclusions of our work.

Furthermore, we have submitted all data associated with this work to PANGAEA, as per journal instructions. Thus, we have replaced the headings of the supplementary tables (which were submitted as such awaiting approval of PANGAEA) with the doi's of the associated data table.

I hope you find our revised manuscript suitable for publication in Biogeosciences.
On behalf of all co-authors,

Francien Peterse

R#1:
I'd like to see some discussion of how this system is unique, and whether they think that would bias the results. For example, Chala is a crater lake with very little terrestrial input. What about a system with very different morphology that does get a lot of terrestrial input? This sort of reflection would be easy to add to the implications section, and would figure strongly in how broadly the results from Chala are likely to apply.

*Reply: The input of soil material into a lake will indeed depend on the morphology of the catchment. In addition, local climate may be an important factor, as e.g. precipitation events are needed for soil mobilization and transport to the lake. However, it remains impossible to determine the exact contribution of soil-derived brGDGTs to the signal that is stored in the sediments of a lake: In contrast to rivers and the coastal marine environment, where in situ brGDGT production can be recognized based on the relative abundance of 6-methyl brGDGTs (De Jonge et al., 2014) or the weighed number of rings in the tetramethylated brGDGT (Sinninghe Damsté, 2016), there are currently no indices that systematically indicate aquatic brGDGT production in lakes. Pending the development of such index, so far the source of brGDGTs is mostly inferred from comparisons of brGDGT distributions between lake sediments and catchment soils, but the offsets appear to lack consistency among the studied systems. Currently, the number of lakes studied in enough detail so far is too low to recognize the general patterns that may allow evaluation of whether or not Lake Chala is unique, and why. We have clarified this in our discussion, see lines 818-827.*

*Given the clear indications for in situ production in Lake Chala, but good relation with brGDGT distribution and modeled temperature in the East African Lakes calibration set (Russell et al., 2018), it seems that the brGDGT-based temperature calibration already takes the additional production into account, as we also suggest in our manuscript. A logical next*

*step would be to derive a lake-specific transfer function based on a global dataset rather than East African lakes only.*

I did notice a few typos scattered throughout (e.g. line 199 should read "…liquid chromatograph. . ."; line 648 should read ". . .related to temperature. . ."), but one more detailed read by the authors should find these.

*Reply: We thank the referee for catching these. We have carefully re-read our manuscript and made corrections where appropriate.*

R#2:
Specific Comments:
Lines 37-38: What does (sub-) mean? Are you talking about the tropics or subtropics? Speleothems are in both?

*Reply: We refer to both tropics and subtropics in our text as areas where lake sediments are the most common natural archive of continental climate history. Speleothems are indeed also used for tropical and subtropical climate reconstructions, so we changed the text accordingly.*

Lines 46-47: Why is temperature the most important climate parameter to reconstruct? Particularly if you're focusing on the tropics, temperature doesn't change much.

*Reply: Whereas temperature change in the tropics is indeed relatively modest even on glacial-interglacial time scales (~2-4 °C at sea level; e.g. Loomis et al. 2017; Chevalier et al. 2020), this has nevertheless major impact on tropical continental rainfall through its control on sea-surface evaporation and monsoon dynamics between the ocean and adjacent continents. Therefore, no rainfall or moisture-balance reconstruction from the tropics can be properly interpreted without knowing local/regional temperature history as reference frame. More broadly, a record of past tropical temperature evolution is needed as low-latitude end-member to determine the meridional temperature gradient, and to reconstruct global heat distribution, through time. We have added this motivation to the revised version of our manuscript, see lines 74-82.*

Really nice thorough introduction, very clear and informative. Please include a brief (one sentence) analysis of the samples run in the two different labs. Was there any significant difference in any way? If not, I think just stating that there were no discernable differences would be fine.

*Reply: The instruments in both labs are tuned towards the same standards. Although the sensitivity (and thus detection limit) of the mass spectrometer was slightly different between labs, this has had no influence on the brGDGT-based indices and proxy values. In the revised manuscript we added a sentence to cover this issue, see lines 265-266.*

Be clearer at the end of the discussion section about the drawbacks of applying brGDGTs to downcore studies. What time scales would work best? Could we trust the absolute T values, or focus on variability?

*Reply: As can be seen in our 53-month record of brGDGTs in settling particles (Fig. 7), there is no clear recurrent pattern in both the flux and composition of brGDGTs, and a direct link to*

*temperature is also absent. Interestingly, the flux-weighed average brGDGT signal in the settling particles, as well as that of the sediments, translate into a temperature that is close to the measured temperature using the Russell et al. (2018) East African Lake calibration. This indicates that brGDGTs can be used to reconstruct absolute temperatures for sediments that integrate at least several years. We have clarified this at the end of the discussion section, see lines 836-839.*

I think the authors should consider adding some of the folks named in the acknowledgments to the author list if they contributed a large amount of work, particularly the first three people, which sounds like they did.

*Reply: We thank the referee for the suggestion. However, the contributions to this study made by the people named in the acknowledgements are of a technical (supportive) rather than scientific (interpretative) nature. We feel confident in having properly defined as author those people who made a significant scientific contribution to this work.*

Technical Corrections:

*Reply: Thank you for pointing out these (mostly) textual issues, which are all addressed in the revised manuscript.*

Line 16: SPM is defined, but not used in the rest of the abstract
Line 48: revise 'supposed to be' to 'supposedly'
Line 61: remove comma
Line 63: remove comma after 'additional'
Line 67: remove comma after '(IIIa")'

*Reply: We followed all the above suggestions, or rephrased text for clarity.*

Line 68-71: this sentence is unclear due to the structure and perhaps misplacement of the word 'for' and 'with'. Please revise.

*Reply: Agreed. We revised this sentence as follows: "Furthermore, lacustrine brGDGTs are significantly more 13C-depleted than those in nearby soils, implying that at least some of their sources are distinct (Weber et al., 2015; 2018; Colcord et al., 2017)", see lines 107-109 in the revised version.*

Line 75: add comma after 'Also', but I suggest using a different word, such as 'Further'.
Line 81: add comma after 'soils'
Line 83: add comma after 'factors'
Line 84: separate out light and mixing regime even if they're from the same study.
Line 85: add comma after 'chemistry'
Line 86: add comma after '2010)'

*Reply: All agreed, and changed accordingly.*

Line 103: SPM is defined as suspended particles, but perhaps use suspended particulate matter
Line 190: revise to 'into apolar, neutral and polar fractions'
Line 192: remove 'then' and revise 'similar' to 'similarly'

Line 193: rewrite this sentence perhaps to something like 'The lake sediment samples were also extracted and processed like the SPM'.

*Reply: All agreed, and changed accordingly.*

Line 203: what is Ø?

*Reply: Ø refers to the diameter of the BEH particles in the column.*

Line 423: remove 'is shown in Fig. 8' and reword so that Fig. 8 is just cited
Lines 427 and 519: formatting of R2 >0.2, the spacing is weird
Line 454-456: rephrase or remove the 'on the one hand' and 'on the other'
Line 491: add comma after 'Chala'
Line 515: remove 'but indirect' or add commas around it
Lines 521-522: rephrase to be more direct. 'only a few of the comparisons between. . ... are moderately correlated' or something.
Line 546: change 'less extensive' to something less negative
Line 555: change 'realized' to 'noted'

*Reply: We either adopted all the above suggestions or implemented a different solution to improve clarity.*

Line 559: remove 'also'

*Reply: Here we feel that 'also' must remain, to emphasize that not only the brGDGT distribution in SPM but also that in sediment-trap material ("particles settling through the water column") is dissimilar from that in soils.*

Line 648: change 'temperate' to 'temperature'
Line 666: add 'other' after 'variable'
Line 683: add something like ', which is indirectly related to climate' at the end of the sentence.

*Reply: Agreed, all adopted.*

Formatting inconsistencies:
Include n values with your r2 and p values, particularly when you make a conclusion from the relationship (or lack there of).

*Reply: We have added n values in all such instances, except when the number of samples or observations is already mentioned elsewhere in the same sentence.*

Sometimes the Oxford comma is used (e.g. title of section 2.2.1.), but often it's not. Either is fine, but be consistent.

*Reply: In our opinion, consistent use of the Oxford, or serial, comma does not mean it must be used always or never. We have followed the large majority of British and American style guides (The Times Style Manual, The Guardian Style Guide, The Economist Style Guide, The New York Times stylebook, Fowler's Dictionary of Modern English Usage, etc.) which recommend using it only when useful to avoid ambiguity, most often when the conjunction*

*joins compound terms (e.g. in the 2nd half of section 2.5). We have not used it in simple conjunctions, such as the repeated series of named brGDGT compounds in section 3.3.*

Sometimes supplemental info is cited in the text as 'Table S.2' and sometimes as 'Table S4'. Either is fine, but be consistent.

Sometimes °C follows the temperature value directly, and other times there's a space between. Either is fine (I think?), but be consistent.
Sometimes R2=# and sometimes R2 = #. Either is fine, but be consistent.

*Reply: We have checked the complete manuscript for such inconsistencies and made all required changes.*

Figures:
In general, they look nice, but appear as different people made different plots – I suggest using consistent fonts throughout. Also, the fonts are very, very small. I view the plots as full page graphs, and can still barely see a lot of the axis labels and numbers, particularly in figures 4, 6, and 7. This is crucial to change.

*Reply: We have adjusted the fonts so that they are the same in all figures.*

Fig. 2: Show where in East Africa this is with an inset map.

*Reply: we have added this.*

Fig. 3: The key is confusing because not all the colors are in it. I understand that the descriptions are in the caption, but perhaps have a clearer key with differently weighted lines. For instance, for the water T at various depths have solid lines in different colors (perhaps a bit thinner than what's there), as you do, and then different dashed lines for other variables plotted.

*Reply: We have adjusted this.*

Fig. 4: Font is very small. . . will need to be whole page to barely see it. Perhaps get ride of the y axis labels on the second and third columns, smush them together, and make the axes and colorbar font much larger
Fig. 7: very blurry, but perhaps that's an artifact of the submission system. If not, it'd be great to update it to higher resolution.
Fig. 8: looks really nice!

*Reply: We have made the suggested changes and uploaded the new figures.*

References:
Bodé et al., 2020 Ecohydr. 13, e2171
Buckles et al., 2014 GCA 140, 106-126
De Jonge et al., 2014 GCA 125, 476-491
Guo et al., 2020 Biogeosciences 17, 3183-3202
Miller et al., 2018 Climate of the Past 14, 1653-1667
Russell et al., 2018 Org. Geochem 117, 56-69
Sinninghe Damsté, 2016 GCA 186, 13-21